# Atlas of RNA editing events affecting protein expression in aged and Alzheimer's disease human brain tissue

Yiyi Ma [1], Eric B. Dammer [2,3], Daniel Felsky [4,5], Duc M. Duong[2,3], Hans-Ulrich Klein [1], Charles C. White[6], Maotian Zhou[2,3], Benjamin A. Logsdon [7], Cristin McCabe[6], Jishu Xu[8], Minghui Wang [9,10], Thomas S. Wingo [11,12,13], James J. Lah[11,12], Bin Zhang[9,10], Julie Schneider[8], Mariet Allen[14], Xue Wang[15], Nilüfer Ertekin-Taner[14,16], Nicholas T. Seyfried [2,3,11], Allan I. Levey [11,12], David A. Bennett[8] & Philip L. De Jager [1,6✉]

RNA editing is a feature of RNA maturation resulting in the formation of transcripts whose sequence differs from the genome template. Brain RNA editing may be altered in Alzheimer's disease (AD). Here, we analyzed data from 1,865 brain samples covering 9 brain regions from 1,074 unrelated subjects on a transcriptome-wide scale to identify inter-regional differences in RNA editing. We expand the list of known brain editing events by identifying 58,761 previously unreported events. We note that only a small proportion of these editing events are found at the protein level in our proteome-wide validation effort. We also identified the occurrence of editing events associated with AD dementia, neuropathological measures and longitudinal cognitive decline in: *SYT11*, *MCUR1*, *SOD2*, *ORAI2*, *HSDL2*, *PFKP*, and *GPRC5B*. Thus, we present an extended reference set of brain RNA editing events, identify a subset that are found to be expressed at the protein level, and extend the narrative of transcriptomic perturbation in AD to RNA editing.

[1] Center for Translational & Computational Neuroimmunology, Department of Neurology, Columbia University Medical Center, 630 West 168th street, New York, NY 10032, USA. [2] Department of Biochemistry, Emory University School of Medicine, Atlanta, GA 30322, USA. [3] Integrated Proteomics Core Facility, Emory University School of Medicine, Atlanta, GA 30322, USA. [4] Krembil Centre for Neuroinformatics, Centre for Addiction and Mental Health, Toronto, ON, Canada. [5] Department of Psychiatry, University of Toronto, Toronto, ON, Canada. [6] Cell Circuits Program, Broad Institute, 415 Main street, Cambridge, MA 02142, USA. [7] Sage Bionetworks, 2901 Third Avenue, Suite 330, Seattle, WA 98121, USA. [8] Rush Alzheimer's Disease Center, Department of Neurological Sciences, Rush University Medical Center, Chicago, IL 60612, USA. [9] Department of Genetics and Genomic Sciences, Icahn School of Medicine at Mount Sinai, One Gustave L. Levy Place, New York, NY 10029, USA. [10] Icahn Institute of Genomics and Multiscale Biology, Icahn School of Medicine at Mount Sinai, One Gustave L. Levy Place, New York, NY 10029, USA. [11] Department of Neurology, Emory University School of Medicine, Atlanta, GA 30322, USA. [12] Center for Neurodegenerative Disease, Emory University School of Medicine, Atlanta, GA 30322, USA. [13] Department of Human Genetics, Emory University School of Medicine, Atlanta, GA 30322, USA. [14] Mayo Clinic Florida, Department of Neuroscience, Jacksonville, FL 32224, USA. [15] Mayo Clinic Florida, Department of Health Sciences Research, Jacksonville, FL 32224, USA. [16] Mayo Clinic Florida, Department of Neurology, Jacksonville, FL 32224, USA. ✉email: pld2115@cumc.columbia.edu

RNA editing is a molecular process that introduces another layer of variation in RNA. Since its discovery in the human *APOB* gene in 1987[1], millions of sites have been catalogued as being edited in humans. A study of B cells from 27 individuals identified >10,000 exonic sites[2]; an investigation with six tissues in three healthy individuals catalogued 3,041,422 sites[3]; and the most recent analysis of 8551 samples covering 53 body sites from 552 individuals reported 408,580 RNA editing events across the genome[4].

The most common form of RNA editing is adenosine-to-inosine (A-to-I) editing executed through Adenosine Deaminases Acting on RNA (ADAR) proteins. The three major genes encoding ADAR proteins are *ADAR1*, *ADAR2*, and *ADAR3*. ADAR-related RNA editing has been associated with autoimmune and inflammatory diseases, cancer, and cardiovascular disease[5]. Evidence is mounting that there may be a link between RNA editing and neuropathological traits, and ADAR2 protein has an essential role in murine brain development. In humans, a recent RNA sequencing (RNA-seq) study of human hippocampus samples showed significantly higher gene expression in six Alzheimer's disease (AD) cases compared to six normal controls for *ADAR3*, but not *ADAR1* or *ADAR2*[6]. Also, AD patients have been reported to have significantly lower levels of RNA editing at 14 re-coding sites in 11 genes, where the editing process changes the amino acid sequence of the targets[6]. Half of these sites were discovered previously with targeted sequencing measurements in fewer than 30 AD cases and controls[7].

Although there is a growing set of reference brain RNA editing sites, there has been no full genome-scale study to systematically analyze those events in relation to AD or to assess which pathologic processes (amyloid or Tau) or endophenotypes of AD may be associated with changes in gene editing. Here, we have conducted the largest genome-wide human brain study to date, creating a resource which identifies RNA editing events in 1865 samples across 9 brain regions from 1074 subjects; they are being made available through the AD Knowledge Portal website where our source RNA-seq data is already available (https://adknowledgeportal.synapse.org/Explore/Data). Based on our unique data which include multiple brain regions and proteomes from the same subjects, we are able to study the mapping of RNA editing across different brain regions as well as the extent to which coding events affect the proteome. In addition, with our sample size of 1074 independent subjects (Table S1) with brain tissue data, we return significant results from our rigorous statistical methodology that produced our association signals for AD (Fig. 1).

## Results

**Identification of RNA editing events across different datasets.** The majority of RNA editing events are the canonical A-to-I editing types, which are shown as the A-to-G and T-to-C editing types (≥90%) and the C-to-T and G-to-A types (5%). (Fig. S1). We have identified 112,779 frequent A-to-I RNA editing events, which are defined as those events with frequency ≥10%, across the ten datasets, and 58,761 (52%) of them are not reported before (Fig. 2a). There are an equal number of reported and not-reported-before exonic editing events (Fig. 2b). For each editing event, there is no correlation between its frequency and average editing level (% alternative allele) (Fig. 2c). The majority of the known editing events are specific to brain tissue (Fig. S2).

**Distributions and comparisons of frequent RNA editing events across brain regions.** We analyzed regional differences within each dataset separately because of their heterogeneities in RNA-seq metrics (Table S1); for example, the MAYO CBE dataset had a significantly greater number of total reads, aligned reads, uniquely aligned reads, % of ribosome bases, and greater median 3′ bias than the other datasets. The majority of the frequent RNA editing events exist in ≥2 brain regions (Fig. 3a) (65%, 72%, and 40% for the Religious Orders Study or the Rush Memory and Aging Project (ROSMAP), Mount Sinai Brain Bank (MSBB), and Mayo clinic RNAseq study (MAYO) datasets). We define the

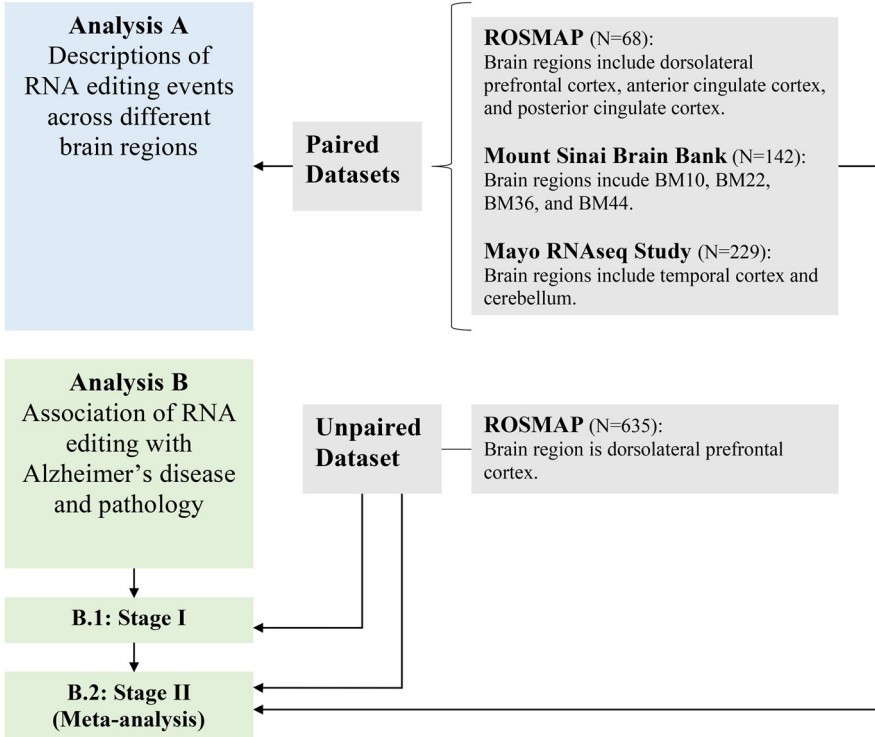

**Fig. 1 Flow chart of our analyses.** This diagram outlines the different datasets used in our report and the different analyses that were performed.

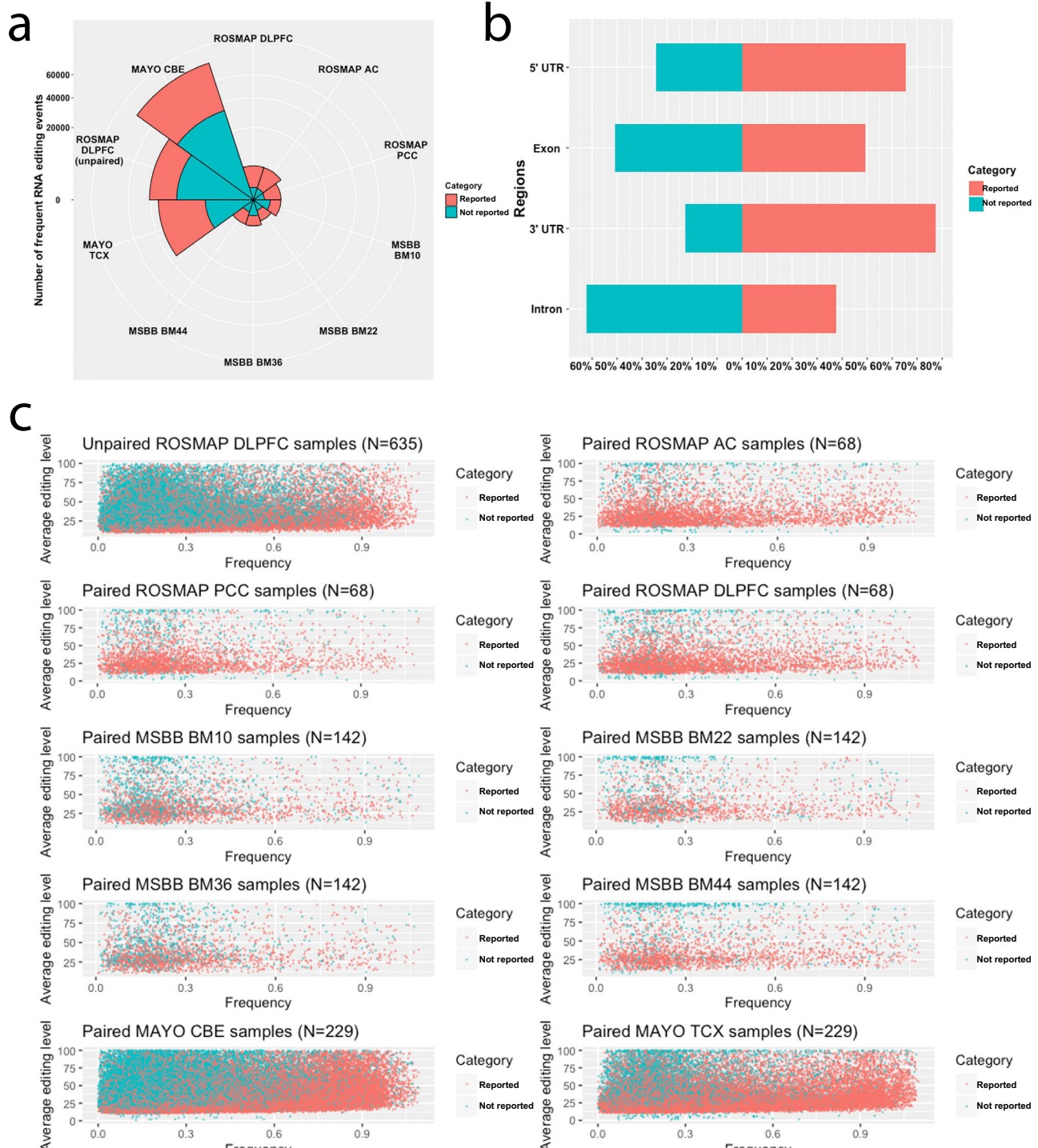

**Fig. 2 Comparisons of the reported and not reported frequent RNA editing events. a** Nightingale plot presents the number of "not reported" and "reported" frequent editing events and their relative proportions within each dataset. **b** Regional distributions of frequent "not reported" and "reported" RNA editing events. **c** Scatter plots of the average level (% of alternative reads over all reads) vs. their frequency of "not reported" and "reported" RNA editing events within each dataset. The color coding for all the plots are red for "reported" and blue for "not reported" frequent RNA editing event.

RNA editing level as the ratio of the edited allele to the total (reference + edited allele) allele count, and most subjects have an overall RNA editing level (the average editing level across all of the identified frequent RNA editing events within each sample from each subject) that is lower than 50% (Fig. 3b). The mean RNA editing level per individual is 37.5% for ROSMAP subjects, 42% for MSBB subjects, and 38% for MAYO subjects. However, there may be variation in the average editing level among brain

regions, as we see (1) a greater editing rate in the posterior cingulate cortex (PCC) than in the anterior cingulate cortex (AC) and dorsolateral prefrontal cortex (DLPFC) in the ROSMAP multi-region study, (2) a reduced editing rate in BM36 than in the other 3 brain regions (BM10, 22, and 44) in the MSBB study, and (3) a greater level of editing in cerebellum (CBE) than in the temporal cortex (TCX) in the MAYO dataset. In terms of the total number of frequent RNA editing events called within each

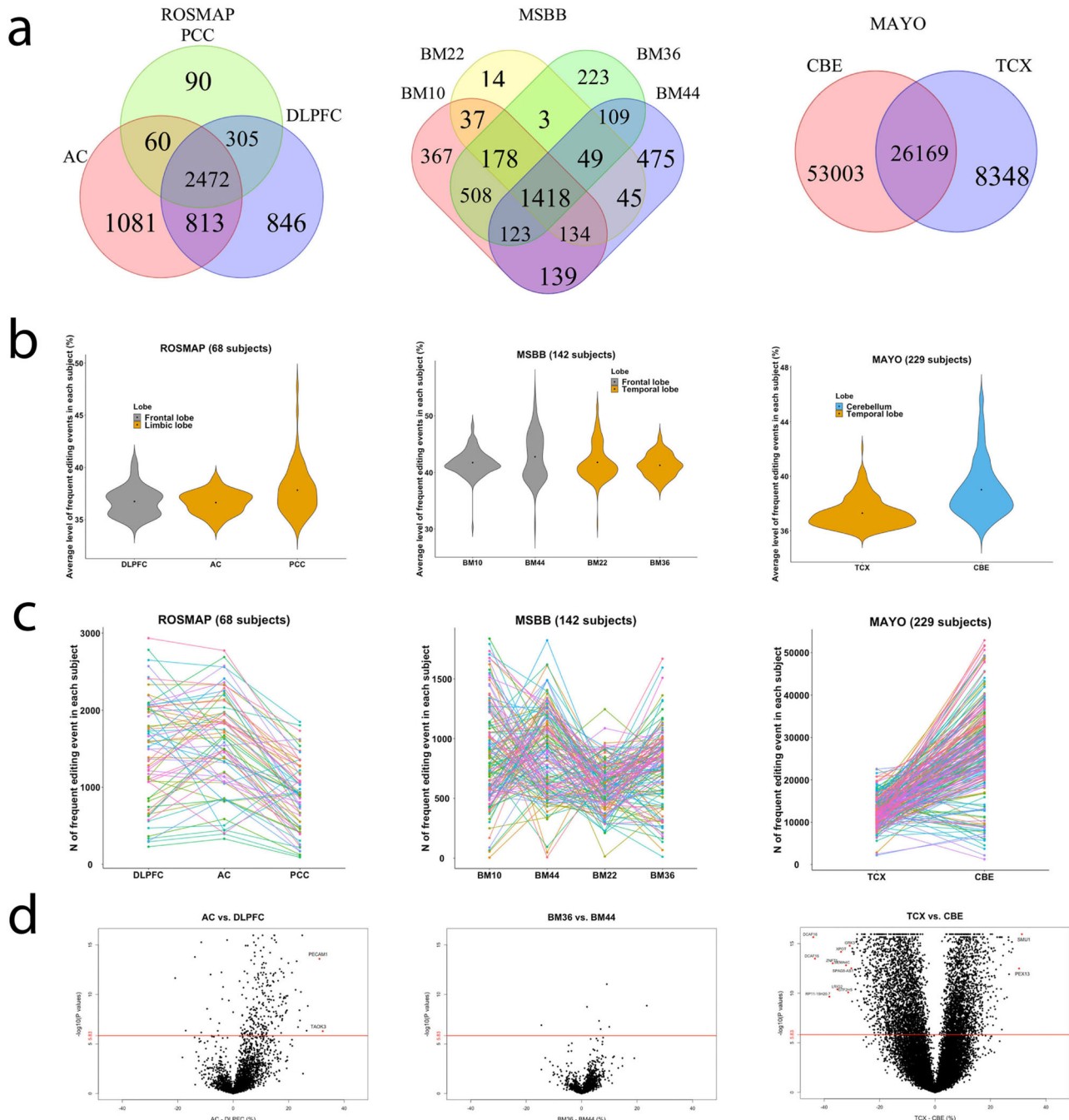

**Fig. 3 Distributions of frequent RNA editing events across brain regions. a** Venn diagrams show the number of frequent (≥10%) RNA editing events present in different regions within different studies. **b** Violin plots show the distribution of the per sample values of the average level of all frequent RNA editing events. **c** The total number of frequent RNA editing events called within each subject across different regions. Each line represents one subject while the connection dots represent the number of frequent RNA editing events called in that subject for the corresponding region. **d** Volcano plots show the differences between 2 tested brain regions in the level (% alternative reads) of each RNA editing event (shown as one dot). X and Y axes display the regression coefficient and its corresponding −log10 transformed P value from the mixed linear regression model with the subject as random effect and fixed covariates of age at death, sex, postmortem interval, and RIN score. The displayed P values were not adjusted by the multiple testings and were derived by two-sided tests. The red horizontal line showed the Bonferroni-corrected P value threshold of 1.49 × 10⁻⁶ (0.05/33,641). The red dots with annotated gene names are those RNA editing events with P ≤ 1.49 × 10⁻⁶ and regression coefficient of the differences between 2 brain regions ≥30%.

subject, there may be also be a region-specific difference, showing, in general, less editing events in PCC, BM22, and TCX compared to the other regions within each of the datasets (ROSMAP, MSBB, and MAYO) (Fig. 3c). This is consistent with the observed differences in *ADAR* gene expression level among these brain regions (Table S3).

We conducted linear mixed models to identify those editing events with a statistically significant difference in editing levels between 2 brain regions within each study after adjusting for biological (age at death, sex) and technical confounding factors (postmortem interval and RIN score). Across the nine paired datasets, we tested the 33,641 frequent editing events that exist in at

least 2 regions, yielding a Bonferroni-corrected significance $P$ value threshold as $\leq 1.49 \times 10^{-6}$ (0.05/33,641). Figures 3d and S2 present the volcano plots illustrating differences in RNA editing levels between 2 regions within each dataset. Some of these are fairly strong; for example, there are two genes (*PECAM1* and *TAOK3*) showing >30% higher editing levels in AC compared to DLPFC, two genes (*ACER3* and *RP11-51713.2*) showing a > 30% reduction in editing levels in PCC compared to AC, two genes (*SMU1* and *PEX13*) showing >30% higher editing levels in TCX compared to cerebellum, and nine genes (*DCAF16*, *GRK3*, *XPOT*, *ZNF71*, *SEMA4C*, *SPAG5-AS1*, *LRIG2*, *GTF2H5*, *RP11-15H20.7*) showing >30% reduction in editing levels in TCX compared to CBE.

**Functional exploration of the non-coding and re-coding RNA editing events**. Altered RNA and protein expression levels have previously been reported as one effect of RNA editing events in the UTR[8,9]. So, we explored two types of functions: (1) the *cis*-effects on the expression levels of the edited genes in transcripts and proteins and (2) the changes in the amino acid sequence for those re-coding RNA editing events, which can alter the coding of an amino acid. In order to remove the potentially inflated correlations between the level of % edited reads and the total reads of the genes/transcripts, we used the binary variable of RNA editing event (yes = 1 and no = 0) for the analysis. We focused on the 635 ROSMAP participants with unpaired data. There were three genes with genome-wide significant *cis*-effects (BETA > 3 and $P < 2.8 \times 10^{-6}$), which are *ORAI2*, *APOL1*, and *PSMD12* (Fig. 4a, left). At the isoform level, there are even more significant hits (absolute BETA > 3 and $P < 1.4 \times 10^{-7}$) (Fig. 4b, left). However, none of the RNA editing events passed the genome-wide significance threshold ($P < 1.8 \times 10^{-6}$) for their association with protein expression for the protein that the target gene encodes (Fig. 4c, left). In addition, compared to the non-coding RNA editing events, the re-coding ones have weaker effects on the expression levels of genes and transcripts although the proportion of significant hits are similar (Fig. 4a–c, middle). A more detailed analyses of the non-coding events located within different genomic regions showed that the *cis*-effects on the expression of the isoforms and proteins were similar. But on the level of the *cis*-effects on the expression of the gene mRNA, the intronic non-coding editing events have less nominally significant effects compared to those events located in the 5′UTR, exons and 3′UTR (Fig. 4a–c, right) where the general patterns of the effect directions were positive rather than negative, indicating that the presence of the non-coding RNA editing events at 5′UTR, exons and 3′UTR were more likely to increase the mRNA expression of the genes.

In order to study the function of re-coding RNA editing events which change the amino acid sequence, we searched for the existence of the predicted amino acid sequence in two datasets: (1) the 171 ROSMAP subjects with both RNA-seq and TMT proteome-wide profiles of the same region, and (2) the 201 subjects in the Banner study who have proteomic data only. With the 171 ROSMAP subjects (Fig. 4d, left), we used the RNA-seq dataset to identify 294 re-coding RNA editing events. The majority of these re-coding events ($n = 247$, 84%) are found at low frequency in the RNA data, in less than 10% of our subjects, and only 3 of these low-frequency events were also identified in the ROSMAP proteomic dataset. The remaining 47 re-coding events (16%) are found at higher frequency ($\geq 10\%$) in the RNA data, and 6 out of these 47 events were also identified in the proteomic dataset. More specifically, the 294 re-coding events found in the RNA data are predicted in silico to yield 459 unique tryptic peptides. From the 373 unique low-frequency predicted tryptic peptides, 4 peptides were detected within the proteomic dataset. From the 47 frequent

re-coding events leading to 86 unique predicted peptides, 8 peptides were detected in the proteomic dataset. As a result, there are 13 unique tryptic peptides encoded by 10 RNA re-coding events identified in the proteomic datasets. The detailed annotation, RNA and peptide sequence for these hits are presented in Table S2. Examples of annotated MS/MS spectra for the edited peptides are provided in Fig. S8. With the 201 Banner subjects, we have identified 5 additional peptide sequences which are consistent with our prediction based on the RNA data of ROSMAP subjects. Thus, overall, using a shotgun proteomic approach, we find only a very small proportion of variation in amino acid sequence at the protein level that is derived from re-coded alleles.

**AD-associated RNA editing events**. We at first evaluated the relation of AD and the level of expression of the three *ADAR* genes (Fig. S4) across the 635 unpaired DLPFC ROSMAP samples. We found no change in *ADAR1* expression, but there is lower expression of *ADAR2* ($P = 0.01$) and higher expression of *ADAR3* in AD cases ($P = 0.01$), while mild cognitive impairment (MCI) subjects are in the middle and the cognitively non-impaired controls have the highest expression of *ADAR2* and lowest expression of *ADAR3*, a potential RNA editing inhibitor[4]. For the composite value including all *ADAR*s (*ADAR1* + *ADAR2*-*ADAR3*) as used in prior studies[4], AD patients have the lowest value, while MCI subjects are in the middle and controls have the highest value ($P = 0.03$).

We then checked for evidence of association between the editing levels at individual RNA editing sites and a diagnosis of AD dementia transcriptome-wide. In Stage I of the analysis (Fig. 1), we used the ROSMAP unpaired dataset including the subjects with non-missing values for all variables used in the regression model, and we tested a total of 40,805 RNA editing sites (yielding a genome-wide significant threshold $P \leq 1.2 \times 10^{-6}$). With the ROSMAP unpaired dataset only, no event passed the Bonferroni-corrected genome-wide $P$ value threshold, but there are 55 events which met a suggestive threshold of association with $P$ values $\leq 1 \times 10^{-3}$. These 55 sites were followed up with a Stage II meta-analysis that includes the Stage I ROSMAP unpaired samples as well as samples from three independent sets of subjects: 142 BM44 samples from the MSBB, 229 TCX samples from the MAYO, and 68 DLPFC samples from ROSMAP participants with multi-region ROSMAP data. The BM44 and TCX region data were selected as being the closest available data to DLPFC which is only available in the ROSMAP subjects included in Stage I. The results of this meta-analysis are not inflated ($\lambda = 1.038$ shown in Fig. S5), and there are six loci showing genome-wide significance in all subjects (Fig. 5a): *SYT11* (top event chr1:155851645, meta-$P = 2.94 \times 10^{-10}$), *MCUR1* (top event chr6:13788361, meta-$P = 1.66 \times 10^{-9}$), *SOD2* (top event chr6:160100882, meta-$P = 1.96 \times 10^{-11}$), *ORAI2* (top event chr7:102096952, meta-$P = 4.8 \times 10^{-7}$), *HSDL2* (top event chr9:115237504, meta-$P = 5.49 \times 10^{-7}$), and *PFKP* (top event chr10:3168677, meta-$P = 1.08 \times 10^{-7}$). The regional plot, correlation matrix of different RNA editing events in the region, and the forest plot of the effects in each dataset for these six loci are presented in Figs. 6 and S5. In sex-stratified analyses, there were no significant male-only results (the smaller subset of the data). In females, an additional event in *GPRC5B* was significant (top event chr16:19876365, meta-$P = 2.79 \times 10^{-7}$), and the events in *SYT11*, *MCUR1*, and *SOD2* remained significant. Directionality is fairly consistent, with greater prevalence of the alternative, edited allele in the context of AD.

All of these genome-wide significant editing events are non-coding; none of the re-coding ones meet a threshold of genome-wide significance, although these two types of editing events present similar distributions of $P$ values (Fig. 5b). Focusing on the

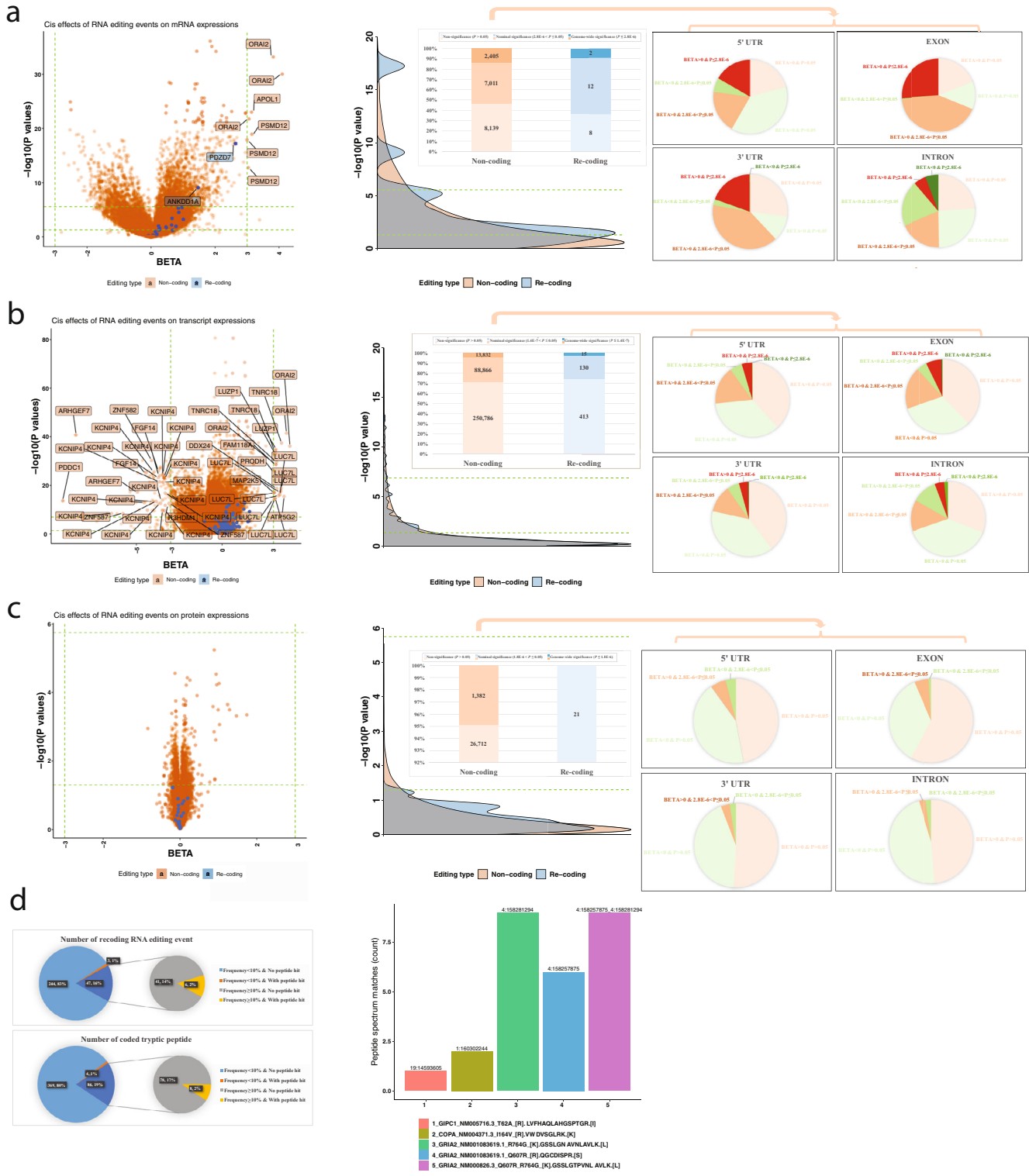

subset of 10 RNA re-coding events which have been matched to 13 edited peptide sequences in the 171 ROSMAP subjects (Table S2), they do not meet our strict threshold of significance for evidence of association to AD, but most have the same effect direction at both the level of RNA and peptide expression relative to AD (Fig. 5c).

All of these editing events, except for *SOD2*, have significant associations with their gene expression in brains (*P* values ranging from $9.17 \times 10^{-23}$ to $8.23 \times 10^{-4}$), and they have

significant associations with the expression of at least one isoform of their corresponding gene (*P* values from $1 \times 10^{-26}$ to $3.69 \times 10^{-2}$). Perhaps more functionally relevant, the level of the *SYT11* editing event has a significant negative association with its protein levels (UNIPROT ID is Q9BT88, *P* = 0.01) (Figs. 5d and 6c), while editing events in *MCUR1* (UNIPROT ID is Q96AQ8, *P* = 0.05) and *GPRC5B* (UNIPROT ID is Q9NZH0.2, *P* = 0.05) have marginal levels of association that deserve further evaluation. The protein level of *ORAI2* was not available, and

**Fig. 4 Functional exploration of RNA editing events (non-coding and re-coding ones).** On the genome-wide scale, we explored the *cis*-effects of RNA editing events on their (**a**) gene expression, (**b**) transcript expression, and (**c**) protein expression. The volcano plots are shown on the left panels, and each dot represents one pair of one RNA editing event (orange for non-coding and blue for re-coding event) and its nearby gene. The *X* axis presents the regression coefficient (BETA) of the exposure variable of RNA editing binary status (0 for no RNA editing and 1 for having RNA editing), and the *Y* axis presents the −log10 transformed *P* values from the generalized linear regression model the fixed covariates of age at death, sex, postmortem interval, and RIN score. The displayed *P* values were not adjusted for the testing of multiple hypotheses and were derived by two-sided tests. The middle panels show the density plots and counts table to summarize the results shown in the volcano plot. The density plots show the comparisons of the association *P* values (−log10) of the non-coding (orange) and re-coding events (blue). The right panels showed the pie charts of the effects of non-coding events located in different genomic regions of 5′UTR (upper left), exons (upper right), 3′UTR (lower left) and introns (lower right). The positive effects are shown in a grade from pink (non-significance, *P* > 0.05) to brown (nominal-significance, Bonferroni-corrected genome-wide significance < *P* ≤ 0.05) to red (genome-wide significance, *P* ≤ Bonferroni-corrected genome-wide significance) while the negative effects are shown in a grade from light green (non-significance, *P* > 0.05) to green (nominal-significance, Bonferroni-corrected genome-wide significance < *P* ≤ 0.05) to dark green (non-significance, *P* > 0.05). **d** Results of a search for non-previously reported peptides based on the observed re-coding RNA editing events in the same subjects in ROSMAP (left) and different subjects in BANNER (right). The left nested pie charts show the matches of RNA re-coding events on the level of transcript (upper panel) and peptide (lower panel). On the right, based on the called 68 RNA re-coding events from unpaired ROSMAP DLPFC samples, five unique peptide sequences exist in an independent dataset with peptide spectrum matches >1.

none of the remaining RNA editing event showed significant association with their corresponding protein levels. We derived 7 principal components (PCs) from the top 7 RNA editing events related to AD. Like the individual editing events, these PCs were also showing significant associations with the expression of genes, isoforms, and proteins (Fig. S7). Overall, we find that several of these AD-associated RNA editing events appear to have downstream effects, manifested as altered RNA levels, isoform levels, and/or protein levels.

Since the level of expression of *ADAR* genes influence the likelihood of editing at a given site, we also evaluated whether these top 7 RNA editing events have significant associations with the expression of one or more of the *ADAR* genes (Fig. 5d). Several of our top RNA editing events show strong relationships with *ADAR* gene expression, for example in *SYT11* with *ADAR2* expression ($P = 1.14 \times 10^{-16}$). This suggests that change in *ADAR* level(s) may be part of the mechanism of the AD-related changes in editing levels.

In secondary analyses, we conducted additional transcriptome-wide evaluations for association with neuropathological traits and aging-related cognitive decline in the discovery dataset of 635 ROSMAP participants with DLPFC data (Fig. 5e) with the genome-wide significant threshold as $P \leq 1.2 \times 10^{-6}$. Four RNA editing events were significantly associated with PHFtau aggregates: *ORAI2* (top event chr7:102096952, $P = 4.72 \times 10^{-8}$), *KCNIP2* (top event chr10:103596067, $P = 6.72 \times 10^{-7}$), *GPRC5B* (top event chr16:19874115, $P = 5.54 \times 10^{-7}$), and *YPEL1* (top event chr22:22078228, $P = 1.77 \times 10^{-7}$). There is one significant RNA editing event associated with the amount of β-amyloid aggregates, which is located in *AC174470.1* (top event chr17:79780692, $P = 9.38 \times 10^{-7}$). There were also two events which showed significant associations with neuritic amyloid plaque burden: *ORAI2* (top event chr7:102096952, $P = 8.47 \times 10^{-7}$) and *CABP1* (top event chr12:121078907, $P = 2.74 \times 10^{-7}$). Finally, two events were significantly associated with the slope of aging-related global cognitive decline: *AC174470.1* (top event chr17:79780692, $P = 5.76 \times 10^{-7}$) and *MUM1* (top event chr19:1371887, $P = 8.50 \times 10^{-7}$).

The top *ORAI2* RNA editing event (chr7: 102096952) is therefore associated with multiple traits (Fig. 6d, e and Table S5), including AD dementia ($P = 4.8 \times 10^{-7}$) and PHFtau accumulation ($P = 4.72 \times 10^{-8}$); it is also close to genome-wide significance with cognitive decline ($P = 3.07 \times 10^{-6}$) and has a more modest association with β-amyloid ($P = 7.31 \times 10^{-4}$). We further found a borderline significant effect of the *ORAI2* editing event on the protein expression of *MAPT* ($P = 0.068$) (Fig. 6f). Thus, we propose that perturbation in RNA editing, in the case of *ORAI2*, is most likely contributing to the accumulation of Tau pathology.

To be thorough, we repeated the analysis for *ORAI2* editing and β-amyloid while accounting for the effect on PHFtau, and the association with β-amyloid is no longer significant, suggesting that it is likely spurious and driven by the correlation between β-amyloid and PHFtau. There was no significant association between those non-A-to-I editing events and the AD-related traits that we have tested (Supplementary Data).

**Discussion**

We have conducted a systematic, transcriptome-wide evaluation of RNA editing in a large multi-region dataset derived from human cortex and cerebellum. With this sample size, we double the current reference of sites where RNA editing takes place in the human brain (available through the Synapse portal: https://www.synapse.org/#!Synapse:syn22335108) and have identified those present in the aging brain. Among the small proportion of sites where the editing event alters the coding sequence of a protein (0.2% of sites), we were able to recover some of the alternative alleles at the protein level in the same individuals as well as in an independent proteomic dataset, extending prior observations that such RNA editing events have a downstream effect. However, most of the proposed re-coding events were not observed at the protein level. Our sample size also allows us to perform an association analysis of RNA editing events in relation to AD and AD-related traits. Importantly, we find reproducible alterations of specific RNA editing events in the context of AD. This is similar to our recent report of reproducible splicing alterations in AD[10] and the broader narrative of specific disruptions in RNA maturation[10,11] and in the epigenome in relation to AD[12]. Our list of top genes associated with AD does not overlap with that from the genetic studies of AD, and none of the RNA editing events located in the AD relevant genes reported by the genetic studies reached a genome-wide significance threshold (Table S6), suggesting that changes in RNA editing in AD are unlikely to be related to genetic risk factors or to affect the same targets. Focusing on the RNA editing component explored in this manuscript, two of the editing events associated with AD dementia stand out: (1) the edited site in the 3′-UTR of *SYT11* which appears to influence SYT11 protein level, and (2) the *ORAI2* 3′-UTR editing event which is associated with multiple diagnostic and intermediate traits, allowing us to propose where, in the causal chain of events leading to AD, RNA editing may have an effect for this gene.

*ORAI2* encodes the calcium release-activated calcium modulator ORAI, which inhibits the function of $Ca^{2+}$ release-activated $Ca^{2+}$ (CRAC) channels by mediating store-operated $Ca^{2+}$ entry (SOCE) in many cell types, including murine naive T cells, neutrophils,

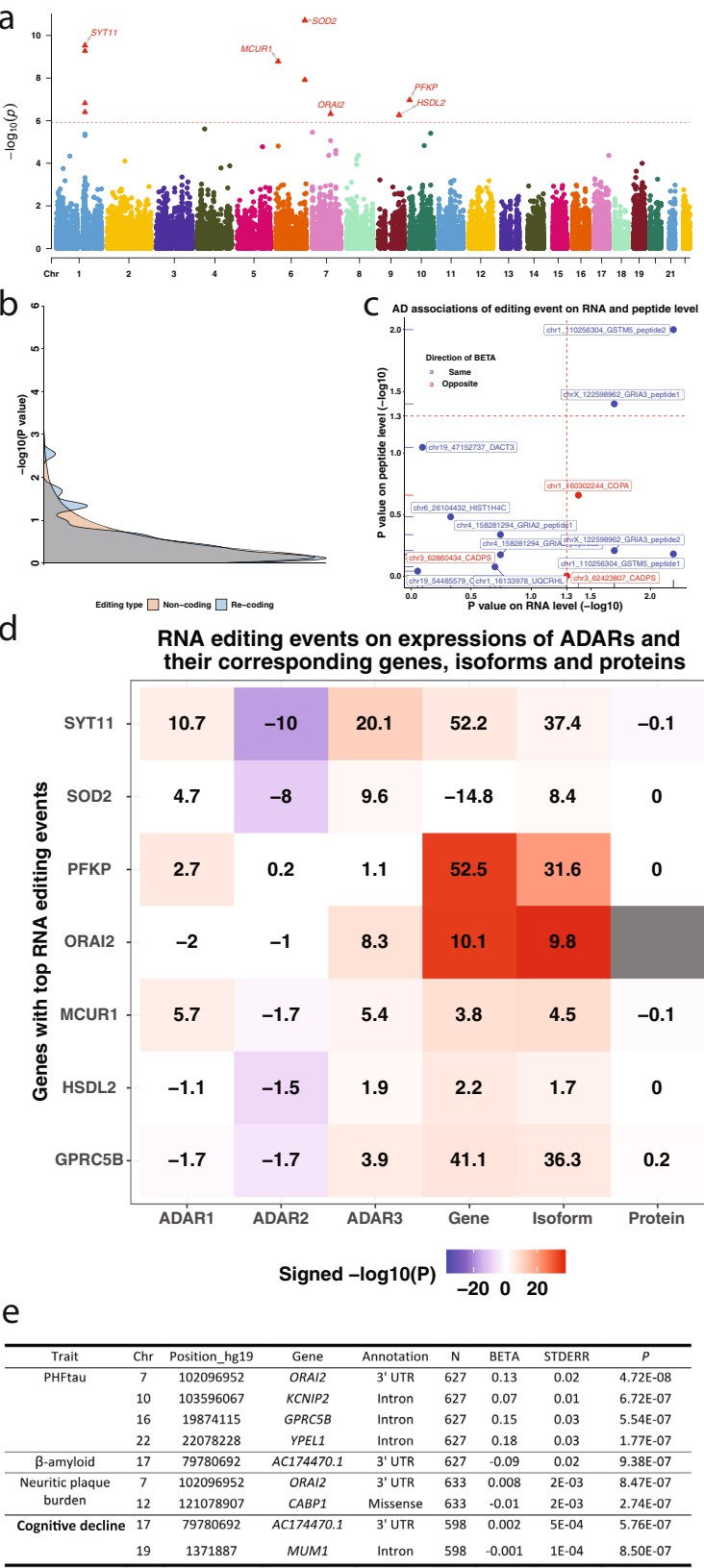

d **RNA editing events on expressions of ADARs and their corresponding genes, isoforms and proteins**

e

| Trait | Chr | Position_hg19 | Gene | Annotation | N | BETA | STDERR | P |
|---|---|---|---|---|---|---|---|---|
| PHFtau | 7 | 102096952 | *ORAI2* | 3' UTR | 627 | 0.13 | 0.02 | 4.72E-08 |
| | 10 | 103596067 | *KCNIP2* | Intron | 627 | 0.07 | 0.01 | 6.72E-07 |
| | 16 | 19874115 | *GPRC5B* | Intron | 627 | 0.15 | 0.03 | 5.54E-07 |
| | 22 | 22078228 | *YPEL1* | Intron | 627 | 0.18 | 0.03 | 1.77E-07 |
| β-amyloid | 17 | 79780692 | *AC174470.1* | 3' UTR | 627 | -0.09 | 0.02 | 9.38E-07 |
| Neuritic plaque burden | 7 | 102096952 | *ORAI2* | 3' UTR | 633 | 0.008 | 2E-03 | 8.47E-07 |
| | 12 | 121078907 | *CABP1* | Missense | 633 | -0.01 | 2E-03 | 2.74E-07 |
| **Cognitive decline** | 17 | 79780692 | *AC174470.1* | 3' UTR | 598 | 0.002 | 5E-04 | 5.76E-07 |
| | 19 | 1371887 | *MUM1* | Intron | 598 | -0.001 | 1E-04 | 8.50E-07 |

astrocytes, dendritic cells, and macrophage as well as human fibroblasts[13]. The neuronal SOCE pathway is important to multiple neurodegenerative diseases such as AD, Parkinson disease, and Huntington disease[14] because of its regulation of synaptic plasticity. Our results emphasize the "Calcium hypothesis" of AD[15] by suggesting that the frequency of RNA editing events in *ORAI2* may shift $Ca^{2+}$ homeostasis in synapses in a way that contributes to the accumulation of PHFtau and the downstream effects on cognitive decline; in vitro work on neuronal function will be necessary to explore these hypotheses.

**Fig. 5 Associations of the top RNA editing events with AD risk and pathologies. a** Manhattan plot of the Stage II association analysis of RNA editing with clinical status of AD. Each dot represents one RNA editing event, and the X and Y axes show its genomic coordinate and −log10 transformed meta-analyzed P value. The horizontal red dashed line shows the Bonferroni-corrected genome-wide significance threshold ($P \leq 1.2 \times 10^{-6}$) and those passing the threshold were shown as the red triangles with gene names. **b** The density plots showed the comparisons of the AD association P values (−log10) of the non-coding (orange) and re-coding events (blue). **c** Scatter plot showed the P values (−log10) of the associations of the re-coding event with Alzheimer's disease based on both the RNA-seq (X axis) and proteomic dataset (Y axis). Blue dot and font represent those events with regression coefficients in the same direction (RNA-seq vs. peptide analysis), while the red dot and font represent those events with regression coefficients in the opposite direction. **d** The matrix plot shows the associations of the top RNA editing events with expressions of ADARs and the mRNA and protein expressions of the gene harboring the editing event. The transformed BETA values (times 100) of effect of RNA editing level (% editing) on the outcomes are presented. The signed −log10(P) values were coded for different colors where white was for values between −1.3 and 1.3 (P = 0.05), darkening blue was for negative values from 0, and darkening red was for positive values from 0. **e** Top RNA editing events associated with AD pathologies in Stage I samples. BETA, STDERR, and P represent the regression coefficient, standard error, and corresponding P values of the generalized linear model of RNA editing level (% alternative allele) as the exposure and each of the trait as the outcomes. The displayed P values were not adjusted by the multiple testings and were derived by two-sided tests.

*SYT11* encodes synaptotagmin XI, which regulates phagocytosis and cytokine secretion in macrophages and interacts with parkin, a protein found in the core of Lewy bodies. The top *SYT11* AD-related editing event we found (chr1: 155851645, hg19) was reported previously[16] in a lymphoblastoid cell line (LCL) with a comparable mean level of editing (% alternative allele) (5.89% in our brain samples vs. 7.9% in LCL). Besides its AD association, we further present evidence that this 3′UTR editing event has a downstream effect on protein abundance, which is in line with the proposed function of 3′-UTR editing events[17]. We found that those subjects carrying the editing event have a significantly lower protein level of SYT11, perhaps because of reduced RNA stability, or tRNA codon bias. Our RNA editing study links *SYT11* to the narrative of altered vesicular formation that has emerged from AD genetic studies, and to potential synapse loss, which is the strongest correlate of dementia[18].

We also identified three mitochondrial-related proteins with AD associations: *SOD2*, *MCUR1*, and *PFKP*. *SOD2* encodes superoxide dismutase 2, a mitochondrial matrix enzyme that scavenges oxygen radicals. The role of *SOD2* in AD is controversial with supportive evidence in animals[19] but not in protein data from humans[20]. The *SOD2* RNA editing event (chr6:160100882, hg19) is significantly associated with AD status in all four datasets (Fig. S5), suggesting its potential role with AD at the RNA level in humans. *MCUR1* encodes mitochondrial calcium uniporter regulator 1, participating in $Ca^{2+}$ flux into mitochondria[21]. We suggest that those subjects carrying the editing event may have lower protein level of MCUR1, with resulting perturbation in mitochondrial function and $Ca^{2+}$ signaling leading to the appearance of AD dementia. *PFKP* encodes the platelet isoform of phosphofructokinase, a key enzyme in glycolysis with high expression in human neurons, and it interacts with a mitochondrial protein, voltage-dependent anion channel 2[22].

A couple of other genes are also significant in our meta-analysis: (1) *HSDL2* which encodes hydroxysteroid dehydrogenase-like 2, a protein involved in fatty acid and lipid metabolism[23] which links it to another pathway implicated in AD susceptibility by human genetic studies. (2) *GPRC5B* encodes member B of the family C of the G protein-coupled receptor superfamily. The neuronal enrichment of the expression is controversial on the mRNA level[24,25], but consistent on the protein level with the highest levels in the neocortex and hippocampus[26]. Murine *Gprc5b* has been shown to impact synaptic formation and neurogenesis[27] and rat *Gprc5b* may be related to microglia activation[28]. In addition, our finding regarding *MUM1* (also known as interferon regulated factor 4, *IRF4*) is noteworthy for its association with cognitive decline. Rats with intracerebroventricular injection of β-amyloid resulted in cognitive impairment and imbalance between *IRF4* and *IRF5*, which was rescued by M2 macrophage transplantation[29]. An amyloid proteinopathy model has also

been reported to harbor microglia with an interferon response[30]. However, evidence supporting a role for interferon responses in human AD has not emerged very strongly so far, although more generic anti-viral responses have been reported[31]. *IRF4* is therefore interesting in this sense, and focuses attention on the interferon pathway in human AD. Our findings may indicate the relevance of microglia to AD pathologies with an intermediary role of RNA editing level.

Although we have a large sample size, we are limited by not having entirely identical brain regions for the meta-analysis (DLPFC, BM44, and TCX); however, these are all neocortical regions. The study may also have a bias towards aged samples. Further, our study still has moderate statistical power, illustrated by the fact that more sites emerge from the meta-analysis. The large number of editing events in the cerebellum, which is essentially devoid of pathology in AD suggests that many of these events may be inconsequential in AD pathology. We also did not adjust for cell-type proportion of the brain tissues in our analyses. Post hoc checking for the top disease-associated RNA editing events, their disease associations were attenuated to some extent but remained significant after adjusting for the estimated proportion of neurons derived from a neuron-specific module of co-expressed cortical genes. This attenuation may be partially driven by the smaller sample size available for this neuron-adjusted analysis or the potential confounding effects of neuron proportion. In addition, total read counts do not affect the results. Our autopsy-based, cross-sectional study design also prevents us from formally demonstrating causality. Furthermore, the mass spectrometry-based proteomic methodologies have technical factors which hinder the complete quantification and identification across samples, and they are subject to ion suppression and interference such that that there is a possibility of no identification of a peptide present in the highly complex input peptide mixture for total brain proteome. This is consistent with the idea that absence of evidence for a peptide in mass spectrometry does not allow the inference or an interpretation that such a result is evidence of absence of that peptide in the cortex. And, the reference proteomic database of 17,112 peptide sequences incorporated the situation when multiple RNA editing events happen at the same time but did not include consideration of genetic variation, as such an inclusion would inflate false discovery due to increasing numbers of decoy peptides and a more intense computational requirement[32]. The low frequency of observation of peptide evidence for coding changes due to RNA editing in ROSMAP cases with paired RNA sequencing may reflect sparse, dissimilar editing events for the specific patient proteomes combined in TMT batches of mass spectrometry-based proteomics. The batch-wise mixtures of individuals in TMT-based quantitative proteomics dilute any peptide found in only one or a small subset of samples within a TMT batch,

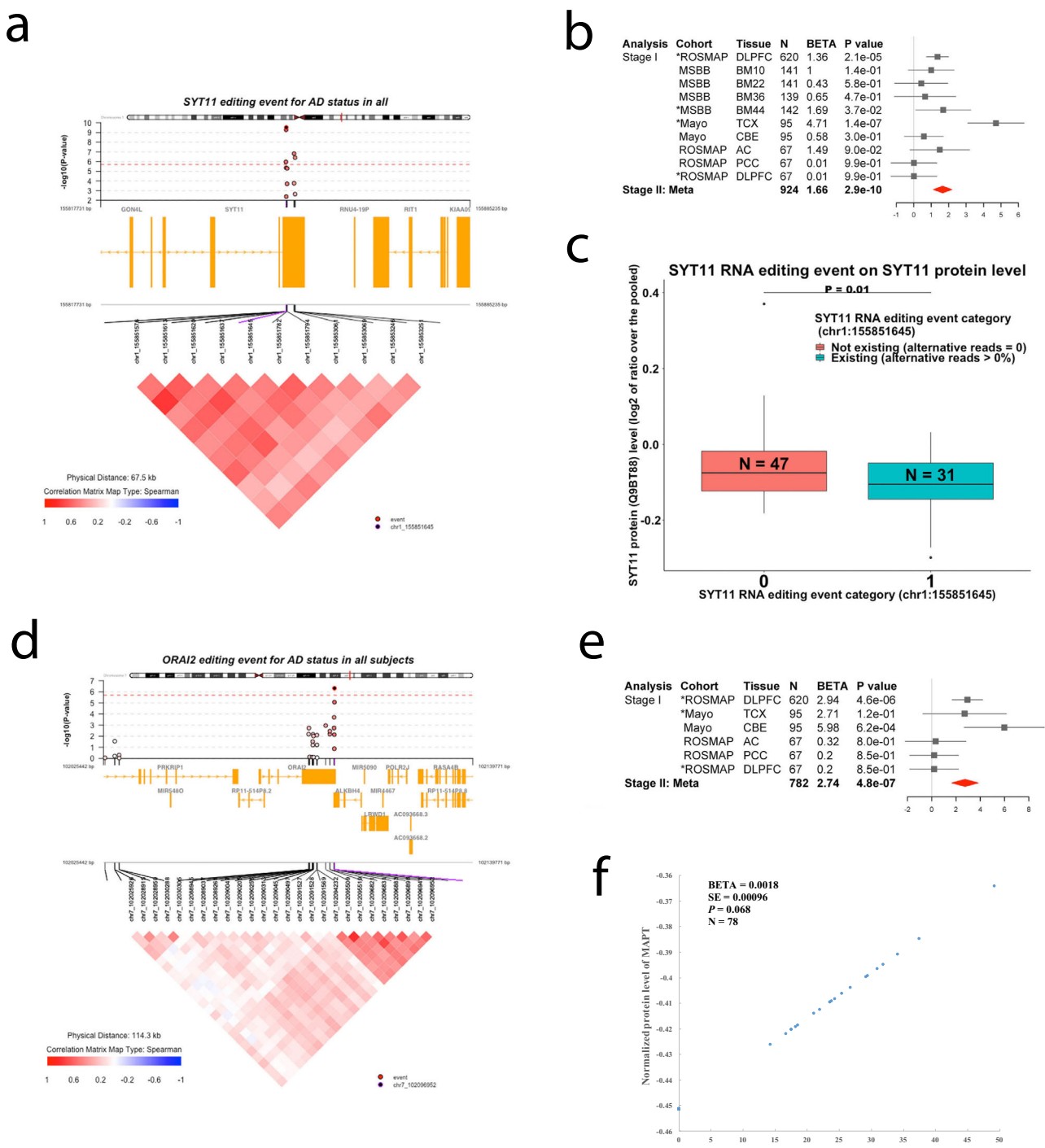

warranting development of additional approaches with better sensitivity for RNA-editing re-coded peptides at the individual level, e.g., selected reaction monitoring. Finally, we elected not to tally, analyze, or comment on the role of infrequent editing events (frequency < 10%) as these are more likely to include sequencing errors.

In conclusion, we have significantly expanded the reference of RNA editing sites where editing occurs in the brain, and we have made this reference available through the AD Knowledge Portal data sharing platform: https://www.synapse.org/#! Synapse:syn22335108. We also identified 7 AD-associated RNA editing events that meet a rigorous threshold of transcriptome-wide significance, and we elaborate the role of two of them: (1)

ORAI2 in the accumulation of Tau pathology that contributes to the cascade of events leading to AD and (2) SYT11 in potential synaptic disruption from the decreased protein levels that may contribute to AD dementia. Our findings need to be replicated and validated in future experiments with model systems.

## Methods

**Study design**. We have assembled RNA sequencing (RNA-seq) datasets derived from 1865 human aging brain samples covering 9 different brain regions from 1074 unrelated subjects. The RNA-seq datasets included in the current study come from the (1) Religious Orders Study or the Rush Memory and Aging Project (ROSMAP) multi-region project, (2) ROSMAP RNA-seq project[33], (3) the Mount Sinai Brain Bank (MSBB) RNA-seq project[34], and (4) the Mayo clinic (MAYO) RNA-seq project[35]. Informed consent was received from all participants or their

**Fig. 6 Regional, forest, functional and pathway plots of SYT11 and ORAI2 RNA editing event. a, d** Regional plot shows the Stage I association results (upper). Each circle represents a frequent RNA editing event with the top one in purple. The correlation (r) between the top one and the others are shown with the color coding of: shallow to dark red for $0 < r \leq 1$, white for $r = 0$, and shallow to dark blue for $-1 \leq r < 0$. The correlation matrix between each pair is shown in a triangle (lower) with the same color coding. (**b, e**) Forest plot shows results in each dataset and Stage II meta-analyzed results. The estimated difference in the mean level of RNA editing (% alternative reads) by clinical AD status (0 for normal controls, 1 for mild cognitive impairment, and 2 for AD) and its 95% confidence interval were illustrated by the filled square and horizontal line for each dataset or the filled red diamonds for the summaries. The displayed $P$ values were not adjusted by the multiple testings and were derived by two-sided tests. **c** Association between RNA editing event with protein expression at *SYT11*. The distributions of RNA editing level by different groups are represented by boxplots where the minimum and the maximum values are represented by the lowest and highest end of the vertical line passing the center of the box, and the first quartile (25%), the median (50%), and the third quartile (75%) of the values are represented as the lower, center, and higher horizontal lines of the box. **f** Association between RNA editing event with protein expression at *MAPT* was represented by the scatter plot where each dot represents one subject included in the analysis. The *X* axis represents the RNA editing level at *ORAI2* (% alternative reads out of the total reads) and the *Y* axis represents the predicted normalized protein level of MAPT by the generalized linear regression line of the effect of the *ORAI2* RNA editing (exposure variable) on the normalized protein level of MAPT (outcome variable) with the adjustments of the covariates of age at death, sex, postmortem interval and RIN score. The displayed $P$ values were not adjusted by the multiple testings and were derived by two-sided tests. Abbreviations: *N*, number of subjects; BETA, regression coefficient; SE, standard error; *P*, *P* value; AD, Alzheimer's disease.

representatives, and sample collections and data processing was approved by the institutional review board (IRB) of each cohort. This study was approved by the IRB of Columbia University (AAAR4962). All the donors of the human tissue used in this study provided informed consent where they agreed to donate their postmortem brains and antemortem clinical data related to the study.

ROSMAP: In brief[33], dementia-free subjects were enrolled into the study with detailed longitudinal measurements of cognitive functions and their postmortem brains were collected for measurements of neuropathologies and molecular omics including RNA-seq and whole-genome sequencing (WGS). ROSMAP RNA-seq project includes postmortem dorsal-prefrontal cortex (DLPFC) samples from 635 unrelated participants in the ROSMAP longitudinal study. ROSMAP multi-region project includes 68 unrelated subjects with samples from all three regions, DLPFC, anterior cingulate (AC), and posterior cingulate (PCC). MSBB: Briefly[34], the Mount Sinai/JJ Peters VA Medical Center Brain Bank (MSBB-Mount Sinai NIH Neurobiobank) cohort included/excluded subjects with stringent criteria to represent the full spectrum of cognitive and neuropathological disease severity in the absence of discernable non-AD neuropathology. The study holds over 2040 well-characterized brains, which were profiled with Brodmann areas. For the current study, MSBB RNA-seq project[34] includes 142 subjects with samples from all four regions: BM10 (the frontal pole), BM22 (the superior temporal gyrus), BM36 (the parahippocampal gyrus), and BM44 (the inferior frontal gyrus), which were shown to be the top 4 most vulnerable regions to AD[34]. MAYO: The Mayo RNA-seq Study[35–37] includes North American Caucasian subjects with neuropathological diagnosis of AD, progressive supranuclear palsy (PSP), pathologic aging (PA), or elderly controls without neurodegenerative diseases, who have donated in total 278 cerebellar cortex (CER; 86 AD, 84 PSP, 28 PA, and 80 controls) and 278 temporal cortex samples (TCX; 84 AD, 84 PSP, 30 PA, and 80 controls), of which 238 were from the same donor, while the remaining had only one of the two tissue region samples. All AD and PSP subjects were from the Mayo Clinic Brain Bank, and all PA subjects were obtained from the Banner Sun Health Institute. Thirty-four control CER and 31 control TCX samples were from the Mayo Clinic Brain Bank, and the remaining control tissue was from the Banner Sun Health Institute.

### Definition of clinical status of AD and neuroROSMAP: pathologic measurements

*ROSMAP.* At every assessment, the clinical diagnosis of cognitive status is determined in a three-stage process (computer scoring of cognitive tests, clinical judgment by a neuropsychologist, and diagnostic classification by a clinician) based on a uniform, structured, clinical evaluation including a battery of 19 cognitive tests per subject. Clinical diagnosis of dementia and clinical AD are based on criteria of the joint working group of the National Institute of Neurological and Communicative Disorders and Stroke and the Alzheimer's Disease and Related Disorders Association (NINCDS/ADRDA). The diagnosis of AD requires evidence of a meaningful decline in cognitive function relative to a previous level of performance with impairment in memory and at least one other area of cognition. Diagnosis of MCI is rendered for persons who are judged to have cognitive impairment by the neuropsychologist but are judged to not meet criteria for dementia by the clinician. Persons without dementia or MCI are categorized as having no cognitive impairment (NCI).

*MSBB.* Clinical dementia rating scale (CDR) was conducted for assessment of dementia and cognitive status for the 6 months preceding death in a multi-step consensus-dependent approach to derive the following scores: CDR = 0 for no cognitive deficits, CDR = 0.5 for questionable dementia, CDR = 1.0 for mild dementia, CDR = 2.0 for moderate dementia, and CDR = 3.0–5.0 for severe to terminal dementia. The longitudinal neuropsychological assessment results were also considered if available in deriving the final consensus CDR score. For the current study, subjects with CDR = 0 are normal controls, CDR = 0.5 are MCI, and CDR ≥ 1 are AD patients.

*MAYO.* The diagnosis of AD is with definite diagnosis according to the NINCDS-ADRDA criteria and had Braak NFT stage of IV or greater, while the normal controls had Braak NFT stage of III or less, CERAD neuritic and cortical plaque densities of 0 (none) or 1 (sparse) and lacked any of the pathologic diagnosis of AD, Parkinson's disease, Lewy body dementia, vascular dementia, PSP, motor neuron disease, corticobasal degeneration, Pick's disease, Huntington's disease, frontotemporal lobar degeneration, hippocampal sclerosis, or dementia lacking distinctive histology[35]. The PA and PSP subjects were excluded from the analysis of the disease associations in the current study.

### Human brain tissue preparation

*ROSMAP.* One hemisphere is cut into coronal slabs and frozen; the other hemisphere is fixed in 4% Paraformaldehyde[33]. Approximately 100 mg of frozen DLPFC were sectioned while still frozen and shipped on dry ice overnight from the RADC to the Broad Institute, where these sections were partially thawed on ice prior to dissection with a scalpel to separate the gray from the white matter and vasculature, resulting in 50–100 mg of gray matter were used to extract RNA.

*MSBB.* Details were described before[34]. Each whole-brain specimen was divided midsagittally. The left half of the brain was cut into 0.8 cm coronal slabs which were flash frozen and kept at −80 °C. For the 4 brain regions included in the current study (BM10, BM22, BM36, and BM44), procedures of dissection, pulverization and aliquotation were kept −80 °C from the fresh frozen, never-thawed 0.8 cm thick coronal tissue blocks. The dissection used a dry ice cooled reciprocating saw and the pulverization used liquid nitrogen cooled mortar and pestle. The pulverized powder is distributed into 50 mg aliquots for each region, which were barcoded and stored at −80 °C until DNA, RNA or protein isolation.

*MAYO.* All brain regions were obtained from frozen sections[35]. A hemibrain or frozen slabs were partially thawed prior to sectioning. Samples were obtained from both the cerebellar cortex (CER) and superior temporal gyrus (TCX) where both were available or from one of the two available regions. White matter and leptomeninges were sectioned away and gray matter was kept for subsequent RNA extraction.

### RNA extraction

*ROSMAP.* The RNA from the gray matter of the DLPFC samples were extracted using Qiagen's miRNeasy mini kit (cat# 217004) and the RNase-free DNase Set (cat# 79254). The quantity and quality of the extractions were conducted with Nanodrop and Agilent Bioanalyzer. We sequence those samples if their RIN > 5 and RNA quantity >5 μg.

*MSBB.* Total RNA extractions were conducted in two RNA preparation cores in Mount Sinai, and two cores used the same kits and modified protocol. RNeasy Lipid Tissue Mini Kit from Qiagen (cat#74804) were used to extract the total RNA from the brain tissues. A slightly modifiable protocol based on manufacturer's protocol (The RNeasy Lipid Tissue Mini Kit Handbook, Qiagen 104945, 02/2009) was used, and the detailed modifications include: (1) all brain tissues (pulverized) were kept on dry ice before adding QIAzol Lysis Reagent, (2) tissues were suspended in the lysis reagent by vortexing with tubes placed on ice, and (3) the tissues were homogenized using a Tissue Ruptor (Qiagen, cat# 79656) at full speed for 20–30 s.

*MAYO.* RNA were extracted from the brain samples via Trizol/chloroform/ethanol method, followed by DNase and Cleanup of RNA using Qiagen RNeasy Mini Kit and Qiagen RNase-Free DNase Set. The Agilent RNA 6000 Nano Chip (Agilent Technologies, Santa Clara, CA) with the Agilent 2100 Bioanalyzer was used to estimate the quantity and quality of the extracted

RNA from all the samples. Only those samples with RIN ≥ 5.0 were sequenced[35–37].

## RNA-seq library preparation and sequencing

*ROSMAP.* RNA-seq library was prepared by the Genomics Platform at the Broad Institute using the strand-specific dUTP protocol with poly-A selection. This method begins with poly-A selection followed by first strand-specific cDNA synthesis, and then uses dUTP for second strand-specific cDNA synthesis followed by fragmentation and Illumina adapter ligation for library construction. Those samples with RIN > 5 and RNA quantity >5 μg were followed up with sequencing on the Illumina HiSeq with 101 bp paired-end reads. The first 12 samples in the "batch 0" included 2 males and 2 females from each of the three clinical AD status (normal controls, MCI, and AD) and their sequence reached coverage of 150 M reads which act as a deep coverage reference. The remaining samples (batch 1 to 6 and 8) were sequenced with coverage of 50 M reads. Samples with similar RIN were pooled together to construct sequence libraries (from batch 1 to 6) because varying RIN scores leads to a larger spread of insert sizes during library construction and uneven coverage distribution throughout the pool. We noticed that samples with lower RIN scores between 5 and 6 had more adapter contamination. We have a later additional samples (batch 7) and the multi-region project samples were run with a modified low-input-quantity requirement (only 250 ng of input RNA) approach of the Illumina TruSeq method to be strand specific and also with larger insert sizes, which construct a library closely resembles the library obtained by the dUTP method. Sequencing was conducted using the Illumina HiSeq2000 with 101 bp paired-end reads for a target coverage of 50 M paired reads.

*MSBB.* Illumina TruSeq RNA Sample Preparation Kit v2 (Illumina, San Diego, CA) was used to construct RNA-seq library. The enrichment of the coding and long non-coding RNA was achieved by depletion of rRNA from total RNA using the Ribo-Zero rRNA Removal Kit (Human/Mouse/Rat) (Illumina, San Diego, CA). Random hexamer-based synthesized cDNA was end-repaired and ligated with appropriate adaptors for sequencing. The library then underwent size selection and purification using AMPure XP beads (Beckman Coulter, Brea, CA). During PCR amplification, the appropriate Illumina recommended 6 bp barcode bases are introduced at one end of the adaptors before loading onto the sequencer, Bioanalyzer (Agilent, Santa Clara, CA) and Qubit fluorometry (Life Technologies, Grand Island, NY) were used to measure the size and concentration of the libraries. Sequencing was run on the Illumina HiSeq 2500 System with 100 bp single-end reads, according to the standard manufacturer's protocol (Illumina, San Diego, CA).

*MAYO.* The RNA-seq samples were randomized across flowcells, taking into account age at death, sex, RIN, Braak stage, and diagnosis[35,36]. The library construction and sequencing were conducted in the Mayo Clinic MGF Gene Expression Core and Sequencing Core. RNA-seq libraries were prepared using the TruSeq RNA Sample Prep Kit (Illumina, San Diego, CA), and the resulting library concentration and size distribution was determined on an Agilent Bioanalyzer DNA 1000 chip. All samples were run in triplicates using barcoding (3 samples per flowcell lane). Sequencing was conducted on the Illumina HiSeq4000 platform with 101 bp paired-end reads[35].

## Alignment of RNA-seq

*ROSMAP.* For the unpaired 635 individuals with DLPFC samples, we applied trimming at first to trim out those low-quality bases (Q10) from beginning and end of each read and the reads of adapter and rRNA. The alignment to the human reference genome hg19 build was conducted via Tophat (v2.1.1) with a non-gap aligner (Bowtie1). For the ROSMAP paired multi-region RNA-seq project, we followed the same data processing procedures as the RNA-seq reprocessings of MSBB and MAYO described later in order to be consistent. We aligned to the reference genome of GENCODE24(GRCh38) using STAR (v2.3.0e) with "two-passMode" set as "Basic". Picard (v2.17.4) functions of "CollectA-lignmentSummaryMetrics" and "Collect RnaSeqMetrics" were used to collect quality metrics and sequencing covariates for each sample.

*MSBB and MAYO.* The MSBB and MAYO RNA-seq reads were initially aligned separately with different tools against different human genome references builds. The raw sequence reads of MSBB RNA-seq datasets were aligned to human genome hg19 with the STAR aligner (v2.3.0e), while the MAYO RNA-seq reads were aligned using the SNAPR software, an RNA sequence aligner based on SNAP, using GRCh38 reference and transcriptome GRCh38.77. SNAPR filters fastq reads by Phred score (>80% of the read must have a Phred score > = 20) and simultaneously aligns each read (or read pair) to both the reference genome of GRCh38 and transcriptome of GRCh38.77. Alignment with SNAPR starts with the creation of hash indices built from both a reference genome GRCh38 and transcriptome

GRCh38.77. The best alignment is written to a sorted BAM file with read counts simultaneously tabulated and written for each sample. Read counts are given by gene ID and transcript ID (two separate files). Mayo Clinic RNA-seq data was previously tested for the read counts generated by SNAPR vs. the read counts generated by HT-Seq and found to be very comparable. Considering the comparability, we downloaded the cross-consortia reprocessed RNA-seq bam files of MSBB (syn8540822) and MAYO (syn8540820 for temporal cortex and syn8540821 for cerebellum). The details of the reprocessing are described on the Synapse platform with Synapse ID of syn9702085. In brief, the aligned BAM files from each study mentioned above were converted back to the FASTQ files using Picard SamToFastq function. Picard SortSam function was used for the MAYO paired-end libraries to ensure the correct order of R1 and R2 in the intermediate SAM file before the conversion back to FASTQ files. For the MSBB single-end library, the converted FASTQ was concatenated to the unmapped reads FASTQ and then underwent sorting. The alignment of the converted FASTQ files were conducted using STAR (v2.3.0e) with "twopassMode" set as "Basic" and the reference genome was GENCODE24 (GRCh38). Picard functions of "CollectAlignmentSummary-Metrics" and "Collect RnaSeqMetrics" were used to collect quality metrics and sequencing covariates for each sample. We downloaded the re-processed BAM files of MSBB (https://www.synapse.org/#!Synapse:syn8540822) and MAYO (temporal cortex: https://www.synapse.org/#!Synapse:syn8540820 and cerebellum: https://www.synapse.org/#!Synapse:syn8540821 RNA-seq projects from the Synapse platform.

## Gene and isoform expression estimate

*ROSMAP multi-region paired RNA-seq dataset.* The transcriptomic gene expressions were estimated by the RSEM (v1.2.31)[38] applied in 10 parallel threads based on the previously described paired-end aligned transcriptome bam files generated using STAR (v2.3.0e)[39] against the reference gene annotations of GENCODE24 (GRCh38). The estimated values of transcripts per million (TPM) were used to represent the gene expression levels.

*MSBB and MAYO RNA-seq.* The gene expressions of the transcriptome for MSBB and MAYO RNA-seq were downloaded from Synapse platform (MSBB: https://www.synapse.org/#!Synapse:syn8691099.1; temporal cortex of MAYO: https://www.synapse.org/#!Synapse:syn8690799.1; and cerebellum of MAYO: https://www.synapse.org/#!Synapse:syn8690904.1). The detailed descriptions of the methods were described online (https://www.synapse.org/#!Synapse:syn17010685. In brief, the gene expression values are represented by the counts of the aligned reads regarding to each gene based on the reference of GENCODE24 (GRCh38) (https://www.gencodegenes.org/human/release_24.html) using STAR by setting "quantMode" as "GeneCounts".

*ROSMAP unpaired RNA-seq dataset.* We estimate the expression levels of genes and isoforms using RSEM (v1.2.31)[38] according to the reference gene annotation of GEN-CODE v14 in hg19 build of human genome reference (https://www.gencodegenes.org/human/release_14.html). Fragments Per Kilobase of transcript per Million mapped reads (FPKM) values were the final output of our RNA-seq pipeline. Normalizations of the gene expression levels were conducted using quantile normalization within each batch followed with Combat normalization with only "Batch" covariate to remove batch effect (https://www.bu.edu/jlab/wp-assets/ComBat/Abstract.html). The unadjusted gene expressions of ADARs (ADAR1, ADAR2, and ADAR3) were analyzed for their associations with clinical AD status (normal controls = 0, MCI = 1, and AD = 2) using the general linear regression models. For the top RNA editing events, we further analyzed their associations with ADARs and their annotated gene and isoforms by the general linear regression model with the lever of the top RNA editing events (% alternative reads) were treated as exposure variable and the ADARs gene expression and the gene/isoform expression of their annotated gene were treated as dependent variable with the covariates of age at death, sex, RIN score, postmortem interval, study (ROS or MAP). The total available sample size of the normalized gene expression files is 635 while it is 542 for the file of isoform expression. The normalized FPKM values were transformed by taking $\log_2$ values for the gene expression file and those subjects with $\log_2$(FPKM) values > 10 standard deviations were set as 0. In order to focus on those isoforms with robust expression values, we removed from the analysis of those isoforms with normalized FPKM > 0 values in <100 subjects.

## RNA editing event calling.

1. ROSMAP unpaired DLPFC dataset: The RNA editing events were called by UnifiedGenotype tool from the Genome Analysis Toolkit (GATK) (v3.6) based on the Tophat (v2.1.1) aligned BAM files. The reference genome sequence and dbSNP were with hg19 build. The option of "ALLOW_N_-CIGAR_READS" was turned on and all the other options are with default settings. Only the editing events with A to I conversion were included.
2. All paired datasets (MSBB, MAYO, and ROSMAP multi-region): We followed the best practice of RNA editing calling pipeline recommended by Broad Institute (https://www.sevenbridges.com/gatk-best-practice-rna-variant-calling/). Start with the STAR aligned BAM files to the GRCh38 reference genome build, we at first used Picard to add read groups, sort, mark duplicates, and create index with functions of

"AddOrReplaceReadGroups" and "MarkDuplicates". Then, we used GATK (v3.6) tool of "SplitNCigarReads" to split reads into exon segments and hard-clip any sequences overhanging into the intronic regions, which is more appropriate to the RNA-seq reads and reduce false variant callings derived from the mis-alignment of the exonic reads to the overhanging intronic regions. In order to solve the differences in MAPQ value meanings between STAR and GATK, we used GATK's "ReassignOneMappingQuality" function to assign those STAR yielded MAPQ of 255 ("unknown") with 60. We did not conduct any Indel realignment. The editing calling is conducted with GATK "HaplotypeCaller" tool with arguments of "dontUseSoftClippedBases" and "stand_call_conf" set as 20. Only the editing events with A to I conversion were included.

**Quality control of RNA editing events**. We have applied three-steps QC procedures: (1) event-level, (2) sample-level, and (3) subject-level.

1. Event-level QC: We have applied posterior filters to filter out those RNA editing events with (1) total reads less than 20, and (2) alternative reads less than 5, and (3) frequency less than 10%, and (4) those overlapping with the DNA variants based on the WGS data across the subjects within ROSMAP, MSBB, and MAYO, where some subjects do not have the RNA-seq data to be involved in the study. Our posterior filters are considered to be more conservative than the recommended filters[40].

2. Sample-level QC: We at first applied study-specific QC metrics within each study. For the ROSMAP multi-region study, we filtered two samples with less than 3 standard deviations from the mean number of total reads and aligned reads across all the samples in the study. For MSBB samples, the details were described previously[34]. In brief, we have removed those suspicious or spurious samples by genetic concordance checks across different types of sequencing data (WGS, WES, and RNA-seq). For duplicated samples, the sample with higher total reads were selected[35,36]. For MAYO RNA-seq project, the samples identified as PCA outliers and sex-mismatches were removed.

3. Subject-level QC: Only selecting those having samples from all different brain regions within each paired dataset.

**Annotation of RNA editing events**. We have lifted over the editing events called within the ROSMAP unpaired dataset to hg38 using the liftOver tool and relevant human chain file (hg19ToHg38.over.chain.gz) downloaded from UCSC genome browser (https://genome.ucsc.edu/cgi-bin/hgLiftOver, and then we merged the result file with all the other editing events files called within all the other datasets into a master file. We used ANNOVAR annotate_variation.pl (downloaded on 16 April 2018) with reference genome sequence of hg38 and gene annotation of ENSEMBLE to annotate the potential functions of called RNA editing events, including where are those editing events located relative to the gene (5′UTR, exon, 1 kb up/down-stream of the transcription start site, 3′UTR, intergenic region, intron, or ncRNA regions), and which is the closest gene(s). We annotate our called RNA editing events as "reported" or "not reported" based on the Rigorously Annotated Database of A-to-I RNA Editing (RADAR) database (version 2 Human)[41] and GTEx publication[4], which were also transformed to hg38 by the same procedure mentioned above using UCSC liftOver tool. The RADAR version 2 Human database included 2,576,277 editing events reported by the 24 peer-reviewed publications of the RNA editing events in humans from 1991 to 2014, and removed ~3000 human genomic single nucleotide polymorphisms (SNPs) from the database. GTEx[4] reported 408,514 RNA editing events existing in 53 body sites from 552 individuals. Those editing events not reported by the above two resources were annotated to be "not reported" ones.

**Proteomic TMT dataset of ROSMAP**

*High-pH off-line fractionation of ROS/MAP brain tissues*. High-pH fractionation was performed as essentially described[42] with slight modification. Dried samples were re-suspended in high-pH loading buffer (0.07% vol/vol NH₄OH; 0.045% vol/vol formic acid, 2% vol/vol acetonitrile) and loaded onto an Agilent ZORBAX 300Extend-C18 column (2.1 mm × 150 mm with 3.5 μm beads). An Agilent 1100 HPLC system was used to carry out the fractionation. Solvent A consisted of 0.0175% (vol/vol) NH₄OH; 0.01125% (vol/vol) formic acid; 2% (vol/vol) acetonitrile and solvent B consisted of 0.0175% (vol/vol) NH₄OH; 0.01125% (vol/vol) formic acid; 90% (vol/vol) acetonitrile. The sample elution was performed by a 58.6 min gradient with a flow rate of 0.4 ml/min. The gradient goes 100% solvent A for 2 min, from 0% to 12% solvent B in 6 min, from 12% to 40% over 28 min, from 40% to 44% in 4 min, from 44% to 60% in 5 min, and then kept 60% solvent B for 13.6 min. A total of 96 individual fractions were collected across the gradient and subsequently pooled by concatenation[43] into 24 fractions and dried to completeness by SpeedVac.

*TMT mass spectrometry of ROSMAP brain tissues*. All fractions were resuspended in equal volume of loading buffer (0.1% formic acid, 0.03% trifluoroacetic acid, 1% acetonitrile) and analyzed by liquid chromatography coupled to mass spectrometry essentially as described[32] with slight modifications. Peptide eluents were separated

on a self-packed C18 (1.9 μm Dr. Maisch, Germany) fused silica column (25 cm × 75 μM internal diameter (ID); New Objective, Woburn, MA) by an Dionex Ulti-Mate 3000 RSLCnano liquid chromatography system (ThermoFisher Scientific) and monitored on an Orbitrap Fusion mass spectrometer (ThermoFisher Scientific). Sample elution was performed over a 180 min gradient with flow rate at 225 nL/min. The gradient goes from 3% to 7% buffer B in 5 min, from 7% to 30% over 140 min, from 30% to 60% in 5 min, 60% to 99% in 2 min, kept at 99% for 8 min and back to 1% for an additional 20 min to equilibrate the column. The mass spectrometer was set to acquire in data-dependent mode using the top speed workflow with a cycle time of 3 s. Each cycle consisted of 1 full scan followed by as many MS/MS (MS2) scans that could fit within the time window. The full scan (MS1) was performed with an *m/z* range of 350–1500 at 120,000 resolution (at 200 *m/z*) with automatic gain control (AGC) set at $4 \times 10^5$ and maximum injection time 50 ms. The most intense ions were selected for higher-energy collision-induced dissociation (HCD) at 38% collision energy with an isolation of 0.7 *m/z*, a resolution of 30,000 and AGC setting of $5 \times 10^4$ and a maximum injection time of 100 ms.

*Database searches and protein quantification*. In all, 1080 raw files for 45 TMT batches with 24 fractions each were analyzed using the Proteome Discoverer suite (version 2.3 ThermoFisher Scientific). MS2 spectra were searched against the UniProt Knowledgebase (UniProtKB) containing both Swiss-Prot and TrEMBL human reference protein sequences (90,411 target sequences downloaded 21 April 2015), plus 245 contaminant proteins. The database was also augmented with 17,112 translated entries for proteoforms from all possible within-proteoform combinations of the RNA editing events called in RNA-seq datasets of ROSMAP DLPFC samples ($n = 635$). The Sequest HT search engine was used and parameters were specified as: fully tryptic specificity, maximum of two missed cleavages, minimum peptide length of 6, fixed modifications for TMT tags on lysine residues and peptide N-termini (+229.162932 Da) and carbamidomethylation of cysteine residues (+57.02146 Da), variable modifications for oxidation of methionine residues (+15.99492 Da), and deamidation of asparagine and glutamine (+0.984 Da), precursor mass tolerance of 20 ppm, and a fragment mass tolerance of 0.05 Da for MS2 spectra collected in the Orbitrap. Percolator was use to filter peptide spectral matches (PSM) and peptides to a false discovery rate (FDR) of less than 1%. Following spectral assignment, peptides were assembled into proteins and were further filtered based on the combined probabilities of their constituent peptides to a final FDR of 1%. In cases of redundancy, shared peptides were assigned to the protein sequence in adherence with the principles of parsimony. Reporter ions were quantified from MS2 scans using an integration tolerance of 20 ppm with the most confident centroid setting. The MS/MS spectra of predicted edited peptides were manually annotated (Fig. S8).

*Removal of batch-specific variance at peptide level*. Normalized abundances for 250,076 peptide modification forms and 229,871 unique peptide sequences from Proteome Discoverer output were considered as an abundance matrix, of which 20 peptides corresponded to 24 editing sites affecting protein coding which were confirmed to be absent in the core database without RNA-edited entries, and manually confirmed to contain the expected variant amino acids per the edited database entries. Batch artifact removal on the full data was performed using an in-house script that implements a median polish algorithm for removing technical variance (e.g., due to tissue collection, cohort, or batch effects) from a two-way complete abundance-sample data table as originally described by Tukey[44]. The implementation in R is fully documented and available via GitHub (https://github.com/edammmer/TAMPOR). The algorithm implements iterations of the below equation.

$$\frac{abundance}{median(ALL\ SAMPLEs)_{batch}} * \frac{grand\ median}{median\left(\left\{\frac{abundance}{median(ALL\ SAMPLEs)_{batch}} | all\ samples\ from\ batch\right\}\right)}$$
(1)

Briefly, Eq. 1 is applied to each peptide TMT normalized abundance measurement across all samples individually where the first term represents batch-wise median-centered abundance, and the second term is a batch-specific normalization factor comprised of the grand median of all batch-specific medians, divided by the appropriate batch-specific median of median-centered abundances. The data matrix is then log₂-transformed, and each log₂(ratio) is adjusted by subtraction of sample (column)-wise median log₂(ratio) for all proteins. Then, ratios are anti-logged and multiplied by the protein (row)-wise geometric mean extracted before Eq. 1 was executed. This process is iterated until convergence. The use of median polish ensures that the reduction of variance is robust to outliers while the overall algorithm preserves biological variance, and outliers, given that batches have been randomized to not confound batch with diagnosis or other biological traits. The above algorithm is applied only to the matrix culled of peptides that have ≥50% missing values across all batches, and all editing-specific peptides regardless of missingness. The first-term denominator was restricted to pooled global internal standard channels ($n = 2$) in each TMT batch, and the second term to all individual case samples (non-internal standard).

*Outlier removal and regression for covariates*. Prior to regression, outlier removal was performed essentially as implemented in Oldham's 'SampleNetworks' v1.06 R

script as previously published[45]. Twenty-one of 360 case samples were flagged as outliers with sample connectivity beyond 3 standard deviations from the mean. Bootstrap regression of the remaining 339-case-sample TMT reporter abundance matrix was performed, explicitly modeling case status category while removing covariation with age at death, gender, and PMI. Given the relative quantitation peptide data we have, it is not possible to calculate edited/total abundance, where summing total abundance from edited and unedited peptides is not correct due to differential ionization efficiency of the distinct peptides. So, we calculated the value of edited/non-edited ratio (Table S2) because relative abundance as a ratio of sample TMT reporter abundance divided by that for the same peptide from internal standard is free of differential ionization, whereas comparison of relative abundance ratios is only possible across samples and not across different peptides, since division of sample peptide abundance by the internal standard peptide abundance abrogates different magnitudes of the signal for different peptides. The protein-level quantitation came from the standard Uniprot reference database entries and it does not incorporate the peptide quantitation of edited variant peptides, which match only to parts of the 17,112 non-Uniprot protein entries in our custom database.

**Proteomic dataset of The Banner Sun Health Research Institute (Banner) study**. This study includes 201 (101 cognitively normal controls and 100 AD cases) postmortem brain tissue samples of the DLPFC from the Banner Sun Health Research Institute's Brain and Body Donation Program. Detailed methodologies are described on the Synapse platform (https://www.synapse.org/#!Synapse:syn9884314 and Supplementary Methods. In brief, the diluted protein homogenate concentration (<2 M urea) from the brain tissues were digested into peptides which were desalted with a Sep-Pak C18 column (Waters) and dried under vacuum. The resulting peptides (2 μg) were resuspended and separated using a self-packed C18 (1.9 μm Dr. Maisch, Germany) fused silica column (25 cm × 75 μM internal diameter; New Objective, Woburn, MA) with a NanoAcquity UHPLC (Waters, Milford, FA), monitored on a Q-Exactive Plus mass spectrometer (ThermoFisher Scientific, San Jose, CA), and eluted with a 120′ gradient at a rate of 400 nl/min The mass spectrometry cycle was programmed to collect one full MS scan (300–1800 $m/z$ range, 1,000,000 AGC, 150 ms maximum ion time at a resolution of 70,000 at $m/z$ 200 in profile model) followed by 10 data-dependent MS/MS scans (2 $m/z$ isolation width, 25% collision energy, 100,000 AGC target, 50 ms maximum ion time at a resolution of 17,500 at $m/z$ 200). MaxQuant v1.5.2.8 with Thermo Foundation 2.0 was used to analyze the raw data for the 201 samples and searched against the same database as described above. Quantitation of proteins was performed using the label-free quantification (LFQ) intensities given by MaxQuant[46].

**Phenotype file preparations**
*ROSMAP*. We have ROSMAP phenotype files through Rush Alzheimer Disease Center (RADC)[33]. MSBB: We removed and remapped the samples from the file "MSBB_clinical.csv" (syn6101474) according to the downloaded Synapse files of "MSBB_RNAseq_covariates.csv" (syn6100548) and "MSBB_RNA-seq.WES.WGS_sample_QC_info.csv" (syn12178047). We transformed the variable of age at death from character to numeric by changing their values of "90+" to 90. We have notified duplicated samples of the same brain region from the same individual but sequenced twice in different batches. For these duplicates, we selected the batch with higher total reads. If the selected batch is with missing value of their RIN score, we assign the non-missing RIN score from the lower read batch to the higher read batch. We only keep subjects who have bam files from all 4 brain regions. In our study, we include 568 bam files from 142 subjects who has passed QC and with corrected phenotype information. MAYO: For each region (cerebellum and temporal cortex), we used the sequencing covariate files and then removed the subjects mentioned in the QC files. For the cerebellum, the sequencing covariate file is "MayoRNAseq_RNAseq_CBE_covariates.csv" (syn5223705) and the QC file is "MayoRNAseq_RNAseq_CBE_QCdetails.txt" (syn6126119). The filesets of the temporal cortex include: "MayoRNAseq_RNAseq_TCX_covariates.csv" (syn3817650) and "MayoRNAseq_RNAseq_TCX_QCdetails.txt" (syn612611) for the sequencing covariate and QC files, respectively. Further, we excluded those subjects with bam files from only one region. We transformed the variable of age at death from character to numeric by changing their values of "90 +" to 90. As are result, we have included 458 samples from 229 subjects into our current study.

**Statistical analysis**
*Regional comparisons of RNA editing events*. In order to compare the potential regional-differences of RNA editing patterns across different brain regions, we have conducted four analyses by: (1) counting the number of the same editing events co-exist in more than 2 regions within the same study; (2) comparing the distribution of the individual-based overall level of all the called frequent RNA editing events across brain regions within the same study, which was calculated by dividing the sum of the % edited reads for all the RNA editing events by the number of editing events called within that individual; (3) comparing the total number of called

frequent RNA editing events within each individual across different brain regions within the same study; (4) comparing the level of each RNA editing event between 2 different brain regions within the same study by running the mixed linear regression models with subject ID as random effect, and covariates of age at death, sex, postmortem interval (PMI), and RIN score using R lme package.

*Associations of RNA editing events with clinical status of AD*. The primary analysis was conducted within each of the 10 dataset, and a general linear model (glm) was utilized to analyze the associations between RNA editing levels (% alternative reads) and AD status (0, 1, and 2 represent normal controls, MCI, and AD patients). For MAYO datasets, there is no subjects with MCI, so the AD status were coded as 0 for normal controls and 2 for AD patients. The covariates adjusted in the model are slightly different across different studies depending on data availability. For the 4 ROSMAP datasets, we adjusted for sex, age at death, PMI, RIN, experimental batch, and study (ROS vs. MAP). For the 4 MSBB datasets, we adjusted for sex, age at death, PMI, RIN, experimental batch, and race. For the 2 MAYO datasets, we adjusted for sex, age at death, PMI (both the actual and imputed values), RIN, and tissue source ("BannerSunHealth_TomBeach" or "MayoBrainBank_Dickson"). At Stage II, we conducted the meta-analysis on the largest cohort including samples from each study of: ROSMAP unpaired DLPFC, the DLPFC dataset of the paired ROSMAP multi-region RNA-seq project, BM44 dataset of the paired MSBB RNA-seq project, and the temporal cortex dataset of the paired MAYO RNA-seq project. The beta estimates and standard errors from each dataset were meta-analyzed using an inverse variance-weighted, fixed-effects approach implemented in METAL (https://genome.sph.umich.edu/wiki/METAL_Documentation).

*Associations of RNA editing events with AD pathologies and cognition decline*. We conducted the analysis in ROSMAP unpaired 635 subjects with DLPFC samples to explore the associations with PHFtau (tangle density) and β-amyloid (overall amyloid level), neuritic plaque burden, and cognition decline. The variables of the PHFtau (tangle density) and β-amyloid (overall amyloid level) is the mean of PHFtau and β-amyloid protein across 8 regions of hippocampus, entorhinal cortex, midfrontal cortex, inferior temporal, angular gyrus, calcarine cortex, anterior cingulate cortex, and superior frontal cortex (4 or more regions area needed to calculate). The PHFtau are identified by molecularly specific immunohistochemistry (antibodies to abnormally phosphorylated Tau protein, AT8) and cortical density (per mm$^2$) is determined using systematic sampling. The β-amyloid protein was identified by molecularly specific immunohistochemistry and quantified by image analysis to obtain a value of percent area of cortex occupied by β-amyloid protein. The neuritic plaque burden is determined by microscopic examination of silver-stained slides from 5 regions of midfrontal cortex, midtemporal cortex, inferior parietal cortex, entorhinal cortex, and hippocampus (4 or more regions area needed to calculate). The count of each region is scaled by dividing by the corresponding standard deviation and the 5 scaled regional measures are then averaged to obtain a summary measure for neuritic plaque burden. The cognition decline variable was defined as the variable of the random slope of global cognition tests. We run the analysis using the generalized linear model with the above mentioned variables as outcomes and the RNA editing levels (% alternative reads) as exposures and covariates of sex, age at death, PMI, RIN, experimental batch, and study (ROS vs. MAP). The genome-wide significance $P$ value threshold is defined as ≤$1.21 \times 10^{-6}$ (0.05/ total 41,254 called frequent RNA editing events).

*Principal component analysis*. The scaled RNA editing levels (%) (mean = 0 and SD = 1) of each of the top 7 AD-related RNA editing events were input to derive 7 PCs using the R packages of "factoextra" (v1.0.7) and "prcomp" (v.3.6.2).

**Reporting summary**. Further information on research design is available in the Nature Research Reporting Summary linked to this article.

## Data availability
All the data generated in this study have been deposited in the AD Knowledge Portal (https://adknowledgeportal.synapse.org) under accession code of syn22335108. The raw data is protected and are not available due to data privacy laws. The AD Knowledge Portal is a platform for accessing data, analyses, and tools generated by the Accelerating Medicines Partnership (AMP-AD) Target Discovery Program and other National Institute on Aging (NIA)-supported programs to enable open-science practices and accelerate translational learning. The data, analyses and tools are shared early in the research cycle without a publication embargo on secondary use. Data is available for general research use according to the following requirements for data access and data attribution (https://adknowledgeportal.synapse.org/DataAccess/Instructions). Databases used for search include: UniProt Knowledgebase (UniProtKB); Human reference genomes of GENCODE24 (GRCh38) (https://www.gencodegenes.org/human/release_24.html) and GENCODE v14 in hg19 build of human genome reference (https://www.gencodegenes.org/human/release_14.html); dbSNP databases were downloaded from the GATK resources (https://gatk.broadinstitute.org/hc/en-us/articles/360035890811-Resource-bundle).

## Code availability

We have made the variant calling pipeline using the RNA-seq bam file open to the public on https://zenodo.org/badge/latestdoi/420196208. The software and tool used in the study include: Tophat (v2.1.1); STAR (v2.3.0e); RSEM (v1.2.31); Picard (v2.17.4); R (v3.6.1); GATK (v3.6); ANNOVAR (downloaded on 16 April 2018); R packages of "factoextra" (v1.0.7) and "prcomp" (v.3.6.2); METAL (https://genome.sph.umich.edu/wiki/METAL_Documentation).

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

## Acknowledgements

We are grateful to the participants in the Religious Order Study, the Memory and Aging Project. Fundings: ROSMAP: This work is supported by the US National Institutes of Health [U01 AG046152, R01 AG043617, R01 AG042210, R01 AG036042, R01 AG036836, R01 AG032990, R01 AG18023, RC2 AG036547, P50 AG016574, U01 ES017155, KL2 RR024151, K25 AG04190601, R01 AG30146, P30 AG10161, R01 AG17917, R01 AG15819, K08 AG034290, P30 AG10161, and R01 AG11101]. MSBB: Gene Expression and Genomic Variants (WES and WGS) for this study was funded through grant U01AG046170 from the NIH/National Institute on Aging (NIA), a component of the AMP-AD Target Discovery and Preclinical Validation Project. Brain tissue collection and characterization was supported by NIH HHSN271201300031C. Other NIA grants relevant to the data included in the current study may include RF1AG054014, RF1AG057440 and R01AG057907. MAYO RNA-seq project: Study data[35] were provided by the following sources: The Mayo Clinic Alzheimers Disease Genetic Studies, led by Dr. Nilüfer Ertekin-Taner and Dr. Steven G. Younkin, Mayo Clinic, Jacksonville, FL, using samples from the Mayo Clinic Study of Aging, the Mayo

Clinic Alzheimers Disease Research Center, and the Mayo Clinic Brain Bank. Data collection was supported through funding by NIA grants P50 AG016574, R01 AG032990, U01 AG046139, R01 AG018023, U01 AG006576, U01 AG006786, R01 AG025711, R01 AG017216, R01 AG003949, and NINDS grant R01 NS080820, CurePSP Foundation, and support from Mayo Foundation. Study data includes samples collected through the Sun Health Research Institute Brain and Body Donation Program of Sun City, Arizona. The Brain and Body Donation Program is supported by the National Institute of Neurological Disorders and Stroke (U24 NS072026 National Brain and Tissue Resource for Parkinsons Disease and Related Disorders), the National Institute on Aging (P30 AG19610 Arizona Alzheimers Disease Core Center), the Arizona Department of Health Services (contract 211002, Arizona Alzheimers Research Center), the Arizona Biomedical Research Commission (contracts 4001, 0011, 05–901 and 1001 to the Arizona Parkinson's Disease Consortium) and the Michael J. Fox Foundation for Parkinsons Research. EMORY proteomic study: Support for this research was provided by funding from the National Institute on Aging (R01AG053960, R01AG057911, R01AG061800), the Accelerating Medicine Partnership for AD (U01AG046161), the Emory Alzheimer's Disease Research Center (P50AG025688), and the NINDS Emory Neuroscience Core (P30NS055077).

## Author contributions

P.L.D. and Y.M. designed study. Y.M. called RNA editing events of all the paired datasets (ROSMAP multi-region study, MSBB and MAYO RNA-seq Study) and conducted quality controls, data analysis, result interpretations, and manuscript writing. P.L.D. supervised and reviewed all the results and manuscript writing. C.M. conducted the RNA-seq experiments and J.X. called RNA editing events and gene expression levels of genes and isoforms based on the RNA-seq samples of 635 unpaired dorsolateral prefrontal cortex in ROSMAP. E.B.D., D.M.D., M.Z., T.S.W., J.J.L., N.T.S., and A.I.L. provided proteomic datasets. B.A.L. provided whole-genome sequencing across ROSMAP, MSBB, and MAYO participating in AMP-AD consortium. M.W. and B.Z. provided guidance to prepare the phenotype files of MSBB. M.A., X.W., and N.E-T. provided consultants of the interpretation of MAYO datasets. J.S., D.A.B., and P.L.D. provided RNA-seq datasets and phenotypes of ROSMAP. D.F., H.U.K., and C.C.W. reviewed the results. All authors read and approved the final manuscript.

## Competing interests

The authors declare no competing interests.
