## [Peer Review File. · Nature Communications]

Reviewers' Comments:

Reviewer #1:

Remarks to the Author:

In the manuscript "RNA editing in the aging and Alzheimer brain: transcriptomic and proteomic perspectives", Ma et al. have conducted a large-scale genome-wide study to identify RNA editing events across several brain regions and over 1,000 individuals from different cohorts of AD patients and controls. Furthermore, by producing proteomic data that partially matched with the transcriptomic analysis, the authors could assess the presence of aminoacidic changes of the RNA editing events in coding regions.

There is great need for understanding the molecular mechanisms involved in the pathogenesis of AD and this study could identify RNA editing events located in the 3'UTR regions of two protein-coding genes associated with AD dementia. Furthermore, association of editing events in some mitochondrial-related genes with AD supports the long-standing hypothesis of mitochondrial dysfunction in AD patients.

However, the manuscript needs to be improved as there are multiple aspects that are not tackled or investigated only at a superficial level and that, if dealt with greater detail, might improve the overall quality of the manuscript. Specifically, role of RNA editing events at non-coding transcripts and overlap with SNPs associated with AD need to be investigated as they might unveil potentially interesting candidates. Furthermore, some claims need to be experimentally validated. I think the paper would be appropriate for publication in Nature Communications if the following aspects are addressed:

Major points:

1. Although a partial strength of the study is the availability of proteomic data to compare the effect of RNA editing events at protein level, I would urge the authors to look also at RNA editing events found in non-coding RNAs. There is already substantial literature in the field that identifies the involvement of lncRNAs in the AD pathogenesis. On the top of my mind I can think at BACE1-AS, a lncRNA that enhances the cleavage of APP by beta secretase and, in turn, production of A β peptide. Also, RNA editing events located in lncRNA genes have been already suggested to alter stability, structure and/or function of the transcript.

Therefore, I think a more thorough study that includes also analyses on the association of RNA editing events at non-coding regions with AD is needed to improve the quality of the manuscript.

2. Can authors provide some analyses regarding the overlap of RNA editing events and SNP already associated with AD? Probably multiple thresholds for the distance between the editing event and the associated SNP need to be taken into consideration. This is potentially extremely interesting as it could shed some light on the functional outcome of polymorphisms previously associated with AD.

3. Current hypotheses regarding the genetic architecture of common diseases envision contribution of both common and rare variants to disease pathogenesis. How about the role of more rare editing events in AD? I think it would be of great interest to lower the threshold from 10% to 5% and/or 1% and perform the analyses again. I would expect more rare editing variants to have a higher penetrance and, in turn, stronger functional impact on the target transcript.

4. The possibility envisioned by the authors that ORAI2 editing may be involved in accumulation of tau pathology needs to be experimentally validated. The authors need to provide a more solid evidence supporting the claim that editing events may cause a change in the cell calcium homeostasis.

Minor:

- The authors identified RNA editing events located in MUM1 locus associated with aging-related cognitive decline. MUM1 is also known as interferon regulated factor 4 (IRF4). Can the authors speculate a bit in the discussion what might be the relevance of this association taking into consideration the role of inflammation in the aging brain?

- It would be important for the authors to provide a definition of the acronym of difference cohorts in the main text and not just in the Methods section. I find this omission particularly confusing for the authorship.

- I recommend the authors to consider the reorganization of Figure 2 as in the main text Figure 2c is mentioned before Figure 2b.

Reviewer #2:

Remarks to the Author:

In this article, the authors have attempted to characterize the global RNA-editing changes associated with AD brain. The authors have used previously published AMP-AD data including their own ROSMAP data and computed RNA editing sites across the brain and then performed associations of those brain-specific RNA-editing events to clinical AD status, AD pathologies and cognitive decline. The authors also used tangential proteomics data in another dataset to validate some of the RNA editing events. While the idea of this project is really great and interesting, the implementation is lack-luster. It seems that the authors wanted to publish this as soon as possible without spending time analyzing the data in-depth. Several shortcomings of the paper -

a) Why was association of differential editing sites with gene/transcript-level expression not explored in this study?

b) Why was the association of differential editing sites with tangential datasets generated by this group like TWAS, GWAS genes, haQTLs, mQTL, Speakeasy clusters, differential-splicing, etc not explored? Much of these tangential datasets were generated by DeJager lab itself in previous papers. Such comparison will give great insights into AD biology.

c) It is well known that number of editing sites is highly correlated with sequencing depth (see Tran et al., Nat Neuosci, 2018). Do the authors see similar correlation and what do they do as different AD cohorts (ROSMAP, MSBB, Mayo) have very different sequencing depths .

d) Can the authors compare the brain-specific RNA-editing sites from published datasets?

e) Why only A-I editing was used. Though A-I editing is the most common, interesting changes could be in other editing sites, given that it is relatively straightforward to analyze the data for that.

f) Is there change in average editing levels with progression of AD, it is unclear from the analysis.

g) In any analysis, p-values directly depend on sample size. Since the authors have hundreds of samples, it is unsurprising that the p-values they report are significant. Moreover, the p-value levels do not imply biological feasibility. The authors should change the figures (like Fig 5C) to report fold-change, beta-values and not signed log₁₀ p-values (corrected or not)

h) Why was an aligner like RASER (Ahn et al., Bioinformatics, 2015) not used instead of GATK recommended STAR. Aligners like RASER are especially tuned for quantifying RNA-editing sites with more precision.

i) It is unclear what steps were taken to ensure that the RNA editing site resulted from a sequence error. The description in the methods section "Quality Control of RNA editing events" does not go into detail regarding any posterior filters which were used to remove RNA editing sites that were probably caused by technical artifacts in sequencing or read mapping. More details in the methods section are needed.

Reviewer #3:

Remarks to the Author:

In this paper, Ma et al describe an innovative and interesting integration of multi-level omic data to report on RNA edited mRNA variants and their relation to peptides in Alzheimer's Disease. This is an enormously well powered study of some of the most detailed tissue collections, and is the first paper of this kind I have seen in the Alzheimer's field. The proteins it highlights are certainly reasonable functional candidates for a role in AD pathophysiology, but I find the technical approach to integration more interesting than the links to AD! I do have some questions as to how the integration was done, but because this is a novel field I think these questions can be addressed with a more technical discussion and some small scale extra analyses, as opposed to new data gathering.

Major points

1) My largest point of interest comes from the non-synonymous ('recoded') variant detection in the mass-spectrometry. Given that the outcomes being modelled come from percentages of reference variant vs edited variant, it would be great to talk a little bit about changes in sequence changing the observability of an individual peptide. A single amino acid change can change retention time and will change m/z – in extreme cases may even put the peptide into a different fraction. How can this be controlled for? Is the gold standard a smaller replication experiment with labelled standards? This would be a great addition to the discussion where it states that most of the recording events were not observed at the protein level – is this biological or could at least some of it be technical?

2) In a similar vein, I would like to see some discussion as to how a lack of observation in MS doesn't necessarily indicate an absence of a peptide – there may be interference with the edited form or a new modification that makes it difficult to observe.

3) I would also like to see some discussion of the way the reference was designed for spectra to peptide matching. I can think of multiple ways to do this and all ways have their advantages and disadvantages. It looks from the materials and methods like extra peptides that contained the edited variant were added to a standard reference. Was consideration made to the presence of other common variants (such as non-synonymous SNPs) in these peptides? For this reason I like the idea of using a personalized RNA-seq derived reference for each sample, but this likely leads to issues of normalization across samples, and greatly increases computing power and analysis time. Collapsing all variants into one single reference likely decreases the number of confidently identified peptides. So, no perfect approach, but would love to hear a little more justification in the text as to exactly how this method was decided on and whether any trials of other method were used to arrive there.

Minor points

1) What about this study enabled the detection of so many new variants? Is it just the increased power from the large number of samples?

2) Is the method of thresholding for calling variants by RNA only now a field accepted method? If so, a reference should be included in results (there is one in the methods). If not, it would be great to take a handful of case from ROSMAP that also have DNA-seq and show that the variants are genuine for the ones that came out as being of strong interest.

3) The distribution of editing events in some of the plots in Fig 4 looks to be bimodal. Is there any relationship to RNA quality / technical effects that produces this bimodality?

4) What is known about RNA edited mRNAs – do they all make it out of the nucleus? I'm looking at the hugely increased number of calls in the CBE and simply wondering if that is a function of the densely packed nuclei there? So there are more unprocessed nuclear mRNAs sampled in that region than the others?

5) Could the biological and technical factors adjusted for in the linear models on page 8 be listed in the results section rather than the methods?

6) For protein level quantification of variant harboring proteins (page 10), was the peptide containing the variant removed from overall protein level quant? Leaving it in may affect the overall protein quant. (If it doesn't, it would be great to show that in a supplement).

**Reviewer(s)' Comments to Author:**

**Reviewer #1:**

**Comments to the Author:**

*In the manuscript "RNA editing in the aging and Alzheimer brain: transcriptomic and proteomic*
*perspectives", Ma et al. have conducted a large-scale genome-wide study to identify RNA*
*editing events across several brain regions and over 1,000 individuals from different cohorts of*
*AD patients and controls. Furthermore, by producing proteomic data that partially matched with*
*the transcriptomic analysis, the authors could assess the presence of amino acidic changes of*
*the RNA editing events in coding regions.*

*There is great need for understanding the molecular mechanisms involved in the pathogenesis*
*of AD and this study could identify RNA editing events located in the 3'UTR regions of two*
*protein-coding genes associated with AD dementia. Furthermore, association of editing events*
*in some mitochondrial-related genes with AD supports the long-standing hypothesis of*
*mitochondrial dysfunction in AD patients.*

*However, the manuscript needs to be improved as there are multiple aspects that are not*
*tackled or investigated only at a superficial level and that, if dealt with greater detail, might*
*improve the overall quality of the manuscript. Specifically, role of RNA editing events at non-*
*coding transcripts and overlap with SNPs associated with AD need to be investigated as they*
*might unveil potentially interesting candidates. Furthermore, some claims need to be*
*experimentally validated. I think the paper would be appropriate for publication in Nature*
*Communications if the following aspects are addressed:*

*Major points:*

*1. Although a partial strength of the study is the availability of proteomic data to compare the*
*effect of RNA editing events at protein level, I would urge the authors to look also at RNA editing*
*events found in non-coding RNAs. There is already substantial literature in the field that*
*identifies the involvement of lncRNAs in the AD pathogenesis. On the top of my mind I can think*
*at BACE1-AS, a lncRNA that enhances the cleavage of APP by beta secretase and, in turn,*
*production of A β peptide. Also, RNA editing events located in lncRNA genes have been already*
*suggested to alter stability, structure and/or function of the transcript.*

*Therefore, I think a more thorough study that includes also analyses on the association of RNA*
*editing events at non-coding regions with AD is needed to improve the quality of the*
*manuscript.*

**Authors' response:**

Thank you for this comment. RNA editing events that occur in lncRNA were included in the
analysis, although we did not specifically discuss them. Out of the total of 112,779 editing
events, there are 17,216 ones located within non-coding RNAs. We also have included a more
thorough analysis that includes the function and AD associations between the non-coding and
re-coding RNA editing events. As described above (page 2) in the response to the editor, we
have made major revision to the manuscript by adding the genome-wide association analysis of
the expressions of genes, isoforms, and proteins. In brief, the re-coding RNA editing events
have weaker effects than the non-coding ones on the expressions of genes and isoforms. In
terms of the AD associations, the manhattan plot (Fig. 5a) showed the results for both types of
RNA editing events (non-coding and re-coding). All the six RNA editing events which passed the
genome-wide significance threshold are the non-coding ones although the density plot of the *P*

values for these two types of editing events are similar. In Fig. 5c, we were showing the AD
associations for the 13 novel peptide sequences (from 10 re-coding events) existed in the 171
ROSMAP subjects with both RNA-seq data and proteomic profiles where the majority have the
same direction of the effect on AD on the level of RNA and protein but only 2 peptides reached
nominal significance ($P < 0.05$) not the genome-wide significance threshold.

*2. Can authors provide some analyses regarding the overlap of RNA editing events and SNP*
*already associated with AD? Probably multiple thresholds for the distance between the editing*
*event and the associated SNP need to be taken into consideration. This is potentially extremely*
*interesting as it could shed some light on the functional outcome of polymorphisms previously*
*associated with AD.*

**Authors' response:**

Thank you for suggesting this analysis. We have made a review of all the genome-wide
significant AD loci reported by the AD genetics community; however, there is no overlap with the
six top AD-associated genes that we report. Furthermore, we have intentionally removed those
events if they are overlapping with DNA variation, i.e. single nucleotide polymorphism (SNP),
since we need to get the true RNA editing events which occur at the level of RNA not DNA.
Further, we have added one analysis to compare the AD associations between those RNA
editing events located on the genes reported to be related to AD or not. We have presented the
result in the Table S5 where we found 113 RNA editing events are located within the AD
relevant genes but their significance to AD is weaker than those editing events in the genes not
reported to be related to AD by the genetic studies. Finally, we have added one statement into
the first paragraph of the Discussion section that "Our list of top genes associated with AD does
not overlap with that from the genetic studies of AD, and none of the RNA editing events located
in the AD relevant genes reported by the genetic studies reached genome-wide significance
threshold (Table S6), suggesting that changes in RNA editing in AD are unlikely to be related to
genetic risk factors or to affect the same targets." (Page 13 and line 287 of the manuscript).

*3. Current hypotheses regarding the genetic architecture of common diseases envision*
*contribution of both common and rare variants to disease pathogenesis. How about the role of*
*more rare editing events in AD? I think it would be of great interest to lower the threshold from*
*10% to 5% and/or 1% and perform the analyses again. I would expect more rare editing variants*
*to have a higher penetrance and, in turn, stronger functional impact on the target transcript.*

**Authors' response:**

This is an interesting question that we considered carefully. In this first report, we elected to
focus on more common editing events for several reasons. First, we were concerned about the
quality of those rare RNA editing events. This is indicated by our proteomic validation for the re-
coding events called based on the RNA-seq datasets. We have an important advantage in that
171 of the unpaired ROSMAP subjects have both RNA-seq data and TMT proteomic data from
the same brain region. However, 99% (244 out of 247) of the rare (frequency < 10%) re-coding
events do not have evidence of being translated: almost none of these predicted peptides are
observed in the TMT data, while that percentage for the frequent re-coding events (frequency >
10%) is 87%. We stated in the Discussion section (page 17 line 375 of the manuscript file) that
"Finally, we elected not to comment on the role of infrequent editing events (frequency <10%)
as these are more likely to include sequencing errors." In addition, our sample size is not
appropriate to provide robust findings for low frequency events (maximum number is 635 while
the minimum number is 68). We utilized a meta-analysis approach for the AD associations
where the summary statistics from each study were derived at first. In this case, the minimum
sample size of 68 is not sufficient to provide a robust result for the RNA editing event with a
frequency <10% since the number of subjects carrying the editing event will be smaller than 5.

That said, there are probably interesting low-frequency events to characterize in more detail, but
this will require a dedicated effort and larger replication sample sets.

*4. The possibility envisioned by the authors that ORAI2 editing may be involved in accumulation*
*of tau pathology needs to be experimentally validated. The authors need to provide a more solid*
*evidence supporting the claim that editing events may cause a change in the cell calcium*
*homeostasis.*

**Authors' response:**

Thank you for proposing the experimental validation which is very interesting but beyond the
scope of the manuscript. We stated in the Conclusion section that "Our findings need to be
replicated and validated in the future experiments with model systems." (Page 17 and line 384
of the manuscript). We respectfully disagree with the reviewer when she or he indicates that the
validated experiment can provide a more solid evidence. Solid evidence has to be grounded in
rigorous statistical analyses that produce robust, reproducible results. This was the goal of our
manuscript, and we accomplished it, laying an important, robust foundation for future work.
Experimental manipulation can be an important manner with which to further explore a solid
observation from human tissue, but it is prone to all of the limitations of model systems,
especially in this case where we are analyzing human cortical tissue in which multiple cell types
are interacting *in vivo*. As we have emphasized in our conclusion section, our study is focused
on an association analysis with the advantage of observing the associations present in the
primary human tissues from free living individuals, and our association results provide
suggestions with which to guide future mechanistic studies which can be conducted in human
cell lines. However, a negative result from such *in vitro* analyses would be uninterpretable since
there is no evidence to expect that the same chromatin conformation or molecular processes
would be present *in vitro*, making *in vitro* experiments of limited utility at this stage: a positive
result would be nice but could have occurred by chance and a negative result would not mean
that the result from human cortex is incorrect. Nonetheless, we do agree that careful
development of a model system would be a natural next step for our investigations.

*Minor:*

*- The authors identified RNA editing events located in MUM1 locus associated with aging-*
*related cognitive decline. MUM1 is also known as interferon regulated factor 4 (IRF4). Can the*
*authors speculate a bit in the discussion what might be the relevance of this association taking*
*into consideration the role of inflammation in the aging brain?*

**Authors' response:**

We have added a statement in the Discussion section to address this point (page 15, line 341 of
the manuscript): "In addition, our finding at MUM1 (also known as interferon regulated factor 4,
IRF4) is noteworthy for its association with cognitive decline. Rats with intracerebroventricular
injection of β -amyloid resulted in cognitive impairment and imbalance between IRF4 and IRF5,
which was rescued by the M2 macrophage transplantation¹. An amyloid proteinopathy model
has also been reported to harbor microglia with an interferon response². However, evidence
supporting a role for interferon responses in human AD has not emerged very strongly so far,
although more generic anti-viral responses have been reported³. IRF4 is therefore interesting in
this sense, and focuses attention on the interferon pathway in human AD."

*- It would be important for the authors to provide a definition of the acronym of difference*
*cohorts in the main text and not just in the Methods section. I find this omission particularly*
*confusing for the authorship.*

**Authors' response:**

We apologize for this neglect, and we have added the definitions in the sections other than
Methods.

- I recommend the authors to consider the reorganization of Figure 2 as in the main text Figure
2c is mentioned before Figure 2b.

**Authors' response:**

We have reorganized Figure 2 to follow the main manuscript.

**Reviewer #2:**

**Comments to the Author:**

*In this article, the authors have attempted to characterize the global RNA-editing changes*
*associated with AD brain. The authors have used previously published AMP-AD data including*
*their own ROSMAP data and computed RNA editing sites across the brain and then performed*
*associations of those brain-specific RNA-editing events to clinical AD status, AD pathologies*
*and cognitive decline. The authors also used tangential proteomics data in another dataset to*
*validate some of the RNA editing events. While the idea of this project is really great and*
*interesting, the implementation is lack-luster. It seems that the authors wanted to publish this as*
*soon as possible without spending time analyzing the data in-depth. Several shortcomings of*
*the paper -*

*a) Why was association of differential editing sites with gene/transcript-level expression not*
*explored in this study?*

**Authors' response:**

Given the length of the manuscript, we initially elected not to include these analyses as we were
more interested in the protein-level results and in the disease associations. However, in
response to this comment, we have now added such analyses to the Fig. 4. Please see our
detailed response to the editor on page 2 of this letter which addresses this comment in detail.

*b) Why was the association of differential editing sites with tangential datasets generated by this*
*group like TWAS, GWAS genes, haQTLs, mQTL, Speakeasy clusters, differential-splicing, etc*
*not explored? Much of these tangential datasets were generated by DeJager lab itself in*
*previous papers. Such comparison will give great insights into AD biology.*

**Authors' response:**

As described above, we have added the analysis of transcriptome-wide association study
(TWAS) and the proteome-wide association study (PWAS) in response to this comment. There
are many potential analyses to perform given the breadth of multi-omic data that we have on
these subjects. We elected to keep a clear narrative, focusing on a subset of important
analyses, with other analyses deferred for later manuscripts. Assembling all of the suggested
analyses into one manuscript would turn what is already a long, dense manuscript into a laundry
list of results that would be difficult to digest for the reader. Since the RNA editing events belong
to the post-transcriptional mechanism, we felt that it was out of the scope of the main theme of
the current study to conduct the analysis with the pre-transcriptional mechanisms such as the
genome-wide association study (GWAS) and the other epigenomic features of DNA methylation
(mQTL) and histone modifications (haQTL). However, we have added an evaluation of the
known AD loci, as described above in a response to reviewer 1 (see **page 13 line 287** in the
manuscript).

*c) It is well known that number of editing sites is highly correlated with sequencing depth (see*

*Tran et al., Nat Neuosci, 2018). Do the authors see similar correlation and what do they do as*
*different AD cohorts (ROSMAP, MSBB, Mayo) have very different sequencing depths .*

**Authors' response:**

We added the correlations between the number of RNA editing events and total reads into the
Table S1, and they were highly correlated to each other for the most datasets. This may explain
in part that the number of RNA editing events called by the MAYO cerebellum dataset was
higher than the other datasets since the read depths of MAYO cerebellum dataset is higher than
the others (Table S1). This is why we did not include this dataset into the meta-analysis. In
addition, we did the post-hoc check of the top AD loci by further adjusting for the total reads and
all the results remain significant. We added the statement into the Discussion section (page 17
line 375 of the manuscript file) that "Finally, we elected not to comment on the role of infrequent
editing events (frequency <10%) as these are more likely to include sequencing errors."

*d) Can the authors compare the brain-specific RNA-editing sites from published datasets?*

**Authors' response:**

We have added Fig. S1 which showed the tissue specificity of the known RNA editing events
across different tissue types. We annotate our RNA editing events as "known" or "novel" based
on the Rigorously Annotated Database of A-to-I RNA Editing (RADAR) database (version 2
Human) (<http://rnaedit.com>) and GTEx publication (Tan MH., et al., Nature, 2017). According to
their Supplementary File 3, there were 3,710 tissue specific RNA editing events, and 273 were
also identified by us. The number one tissue type of these 273 known tissue-specific RNA
editing events belong to brain (105, 38%). We have added this analysis to the Results section
(page 5 line 104 of the manuscript file) that "The majority of the known editing events are
specific to brains (Fig. S1)."

*e) Why only A-I editing was used. Though A-I editing is the most common, interesting changes*
*could be in other editing sites, given that it is relatively straightforward to analyze the data for*
*that.*

**Authors' response:**

The A-I editing is the most common editing type which has been well-studied to have the
functions on changing the amino acid sequence or expression levels of transcripts and proteins.
The other types of RNA editing might be interesting but we may not have the statistical power to
conduct a robust and validated studies on them given our small sample size. This is an
interesting question that can be pursued in future work.

*f) Is there change in average editing levels with progression of AD, it is unclear from the*
*analysis.*

**Authors' response:**

The first paragraph of the Results section of "AD-associated RNA editing events" (page 8 line
179 of the manuscript file) found that the expression levels of Adenosine Deaminases Acting on
RNA (ADAR) were associated with the AD clinical stages. "We at first evaluated the relation of
AD and the level of expression of the three ADAR genes (Fig. S3) across the 635 unpaired
DLPFC ROSMAP samples. We found no change in ADAR1 expression, but there is lower
expression of ADAR2 (P=0.01) and higher expression of ADAR3 in AD cases (P=0.01), while
the mild cognitive impairment (MCI) subjects are in the middle and the cognitively non-impaired
controls have the highest expression of ADAR2 and lowest expression of ADAR3, a potential
RNA editing inhibitor⁴. For the composite value including all ADARs (ADAR1+ADAR2-ADAR3)
as used in prior studies⁴, AD patients have the lowest value, while MCI subjects are in the
middle and controls have the highest value (P=0.03)." Thus, some of the enzymes involved in
RNA editing are modestly differentially expressed in AD, so we expected to see the average
editing level to be lower in AD patients. However, this is not the case; there is no significant

changes in the average editing levels with AD. We have added the statement of our analysis
into the Result section on page 8 line 187 that "However, we did not find evidence of
association between the average editing levels of each subject with progression of AD (Data not
shown)."

*g) In any analysis, p-values directly depend on sample size. Since the authors have hundreds of*
*samples, it is unsurprising that the p-values they report are significant. Moreover, the p-value*
*levels do not imply biological feasibility. The authors should change the figures (like Fig 5C) to*
*report fold-change, beta-values and not signed log10 p-values (corrected or not)*

**Authors' response:**

We have replaced the plots with images using BETA values (**Fig. 5d**).

*h) Why was an aligner like RASER (Ahn et al., Bioinformatics, 2015) not used instead of GATK*
*recommended STAR. Aligners like RASER are especially tuned for quantifying RNA-editing*
*sites with more precision.*

**Authors' response:**

At the RNA editing events discovery stage, we applied the GATK based on TOPHAT2 (not
START) alignment in the ROSMAP unpaired samples from 635 subjects. It was reported that
the mapping precision is similar between TOPHAT 2 and PASER⁵. Also, a similar calling
pipeline which combines GATK and BWA alignment, was utilized to call the RNA editing events
across a variety of human primary tissues and the callings are validated by Sanger sequencing
338⁴. Understanding the challenges of RNA editing callings from the short read RNA-seq data, we
filtered out those RNA editing events with total reads less than 20, alternative reads less than 5,
frequency less than 10%, and those overlapping with the DNA variants based on the whole
genome sequencing data across the subjects within the same consortium. All of these filtering
criteria are more stringent than the proposed posterior filtering criteria to ensure that we
consider only the most robust sites⁶. At the replication stage, we downloaded the official version
of the STAR aligned bam files which were agreed across the AMP-AD consortium from the
Synapse data portal, and we only focused on those significant RNA editing events that had
been called in the discovery stage.

*i) It is unclear what steps were taken to ensure that the RNA editing site resulted from a*
*sequence error. The description in the methods section "Quality Control of RNA editing events"*
*does not go into detail regarding any posterior filters which were used to remove RNA editing*
*sites that were probably caused by technical artifacts in sequencing or read mapping. More*
*details in the methods section are needed.*

**Authors' response:**

Thank you for raising this important point. We have now clarified our pre-processing filters. As
we have mentioned in the above response, we applied the posterior filters to filter out those
RNA editing events with (1) total reads less than 20, and (2) alternative reads less than 5, and
(3) frequency less than 10%, and (4) those overlapping with the DNA variants based on the
whole genome sequencing data across the subjects that were considered. Our posterior filters
are considered to be more conservative compared to the recommended filters by the
researchers in the RNA editing field⁶.

**Reviewer #3:**

**Comments to the Author:**

*In this paper, Ma et al describe an innovative and interesting integration of multi-level omic data*
*to report on RNA edited mRNA variants and their relation to peptides in Alzheimer's Disease.*
*This is an enormously well powered study of some of the most detailed tissue collections, and is*
*the first paper of this kind I have seen in the Alzheimer's field. The proteins it highlights are*
*certainly reasonable functional candidates for a role in AD pathophysiology, but I find the*
*technical approach to integration more interesting than the links to AD! I do have some*
*questions as to how the integration was done, but because this is a novel field I think these*
*questions can be addressed with a more technical discussion and some small scale extra*
*analyses, as opposed to new data gathering.*

**Major points**

*1) My largest point of interest comes from the non-synonymous ('recoded') variant detection in*
*the mass-spectrometry. Given that the outcomes being modelled come from percentages of*
*reference variant vs edited variant, it would be great to talk a little bit about changes in*
*sequence changing the observability of an individual peptide. A single amino acid change can*
*change retention time and will change m/z – in extreme cases may even put the peptide into a*
*different fraction. How can this be controlled for? Is the gold standard a smaller replication*
*experiment with labelled standards? This would be a great addition to the discussion where it*
*states that most of the recording events were not observed at the protein level – is this*
*biological or could at least some of it be technical?*

**Authors' response:**

The reviewer raises an insightful and important point in that technical factors play a large role in
hindering observation of non-canonical variant peptides including ones resulting from RNA
editing. Not only are retention time and m/z changed by a single residue substitution, but also
ionization efficiency, affecting differential quantitation. For this reason, given the relative
quantitation peptide data we have, it is not possible to calculate edited/total abundance, where
summing total abundance from edited and unedited peptides is not correct due to differential
ionization efficiency of the distinct peptides. The ROSMAP TMT data is from mixtures of TMT
multiplexes (batches) and averages the precursor signal from 8 distinct samples in each
multiplex mixture of peptides, further diminishing the chance of sequencing peptides occurring
at a low frequency in the sample population. Further, independent offline prefractionation of
each TMT multiplex batch of peptides (N=45 batches used for our analysis here) can lead to
batch effects, which, when extreme, lead to missing quantitation in batches. Fortunately, when
quantitation is available, batch effects can be addressed, and we have done this in our analysis
using robust median polish of ratio with global internal standard signal within batch and across
batches (Johnson ECB, et al, Nat Med, 2020). Thus, we are benefitting from normalization to
internal standard comprised of the equal mixture of all analyzed homogenates, present twice in
each batch. Relative abundance as a ratio of sample TMT reporter abundance divided by that
for the same peptide from internal standard is free of effects due to differential ionization, but
comparison of relative abundance ratios is only possible across samples and not across
different peptides, since division of sample peptide abundance by the internal standard peptide
abundance abrogates different magnitudes of the signal for different peptides. As the reviewer
points out, only a calibration curve to obtain absolute quantification, e.g., with known amounts of
heavy stable isotope-labeled peptide for each edited and unedited peptide counterpart, spiked-
in to each sample before fractionation would allow precise calculation of the comparable values
for both counterpart peptides and of the percent edited of total. We have added these technical

explanations into the Method section (page 22 line 494 of the manuscript file) that "Given the
relative quantitation peptide data we have, it is not possible to calculate edited/total abundance,
where summing total abundance from edited and unedited peptides is not correct due to
differential ionization efficiency of the distinct peptides. So, we calculated the value of
edited/non-edited ratio because relative abundance as a ratio of sample TMT reporter
abundance divided by that for the same peptide from internal standard is free of effects due to
differential ionization, whereas comparison of relative abundance ratios is only possible across
samples and not across different peptides, since division of sample peptide abundance by the
internal standard peptide abundance abrogates different magnitudes of the signal for different
peptides."

*2) In a similar vein, I would like to see some discussion as to how a lack of observation in MS
doesn't necessarily indicate an absence of a peptide – there may be interference with the edited
form or a new modification that makes it difficult to observe.*

**Authors' response:**

Indeed, it follows from the above listed technical factors hindering complete quantification and
identification across samples and subject to ion suppression and interference, that there is a
possibility of no identification of a peptide present in the highly complex input peptide mixture for
total brain proteome. We now address this comment by explicitly stating that absence of
evidence for a peptide in mass spectrometry does not allow the inference or an interpretation
that such a result is evidence of absence. Please check the highlighted added text in the
Discussion section of the paragraph of limitations on page 16 line 366 of the manuscript file that
"Furthermore, the mass spectrometry based proteomic methodologies have technical factors
which hinder the complete quantification and identification across samples, and they are subject
to ion suppression and interference such that that there is a possibility of no identification of a
peptide present in the highly complex input peptide mixture for total brain proteome. This is
consistent with the idea that absence of evidence for a peptide in mass spectrometry does not
allow the inference or an interpretation that such a result is evidence of absence of that peptide
in the cortex."

*3) I would also like to see some discussion of the way the reference was designed for spectra to
peptide matching. I can think of multiple ways to do this and all ways have their advantages and
disadvantages. It looks from the materials and methods like extra peptides that contained the
edited variant were added to a standard reference. Was consideration made to the presence of
other common variants (such as non-synonymous SNPs) in these peptides? For this reason I
like the idea of using a personalized RNA-seq derived reference for each sample, but this likely
leads to issues of normalization across samples, and greatly increases computing power and
analysis time. Collapsing all variants into one single reference likely decreases the number of
confidently identified peptides. So, no perfect approach, but would love to hear a little more
justification in the text as to exactly how this method was decided on and whether any trials of
other method were used to arrive there.*

**Authors' response:**

The reviewer brings up an important point about combinatorial variation that was partially
addressed by experiment and reference database design. 17,112 separate full-length protein
entries with all possible combinations of non-synonymous RNA editing events were generated
by in silico translation. Thus, if two edits fall within the same tryptic peptide, they would be
detectable by our approach. However, we did not consider other sources of protein sequence
variation such as SNPs in DNA. Therefore, peptides from translation of edited RNA that also
contains a SNP leading to a coding change would be missed. SNPs are specific to the
individual's proteome being analyzed, so that, to detect them, a personalized database
incorporating all SNPs, if not also indels, would be necessary. The issues with the approach,

have been well described and partially addressed for non-multiplexed (label-free) sample LC-
MS/MS raw data with available paired whole exome sequencing (Wingo et al, J Proteome Res,
2017). Namely, with many variants to incorporate into a personal database, the database size
grows and this hampers sensitivity of confident identification of all peptides due to false
discovery control needing to consider more decoy peptides. The process is also much more
computationally demanding, as a separate database search is performed for every set of raw
data for each individual. This perspective is now incorporated into the text on page 17 line 372
of the manuscript file that "And, the reference proteomic database of 17,112 peptide sequences
incorporated the situation when multiple RNA editing events happen at the same time but not
including the considerations of the genetic variation, as such an inclusion would inflate false
discovery due to increasing numbers of decoy peptides and a more intense computational
requirement ⁷."

*Minor points*

*1) What about this study enabled the detection of so many new variants? Is it just the increased*
*power from the large number of samples?*

**Authors' response:**

According to advances in the RNA editing field, the number and the percentage of the novel
editing events we have identified is not that different from other studies. With 1,865 samples
across 9 brain regions from 1,074 independent subjects, "we have identified 112,779 frequent
RNA editing events (frequency $\geq 10\%$), and 58,761 (52%) of them are novel (Fig. 2a)" (page 5
line 100 of the manuscript). Our total number of 112,799 is only a quarter of the total number of
RNA editing events identified in the previous study with 8,551 samples from 53 body sites of the
552 independent subjects (total number = 408,580) (Tan. MH, et al., Nature, 2017). Also, based
on the previously reported 408,580 RNA editing events by Tan et al., we calculated the
percentage of novel ones which were not reported by them. As a result, we found that 52% of
our 112,779 events (58,761) were not reported by the previous study. But on the perspective of
the previous study, 86% of their 408,580 (349,819) events were not reported by us. It may be
possible that larger number of samples simply provide increase power to detect more RNA
editing events. For example, the number of RNA editing events identified with the 635 ROSMAP
unpaired dorsolateral prefrontal cortex (DLPFC) is greater than that with the 68 ROSMAP
DLPFC paired samples. Also, the MAYO cerebellum (CBE) dataset has significantly greater
number of the RNA editing events than the other datasets, which is in line with our findings that
"the MAYO CBE dataset has a significantly greater number of total reads, aligned reads,
uniquely aligned reads, % of ribosome bases and greater median 3' bias than the other
datasets." (page 5 line 109 of the manuscript file). However, it is also obvious that the
"increased power by larger number of samples" cannot fully explain the issue we have seen.
We speculate that multiple factors may also have contributions, including the inter-subject
variation, tissue-specificity, and brain region-specificity. Our study focused on the brain samples
while the previous one collected samples from 53 sites across the body. Our additional analysis
suggested that the RNA editing sites we have reported are enriched in brain specific sites
reported by the previous study and we have added a statement in the Results section on page 5
line 104 of the manuscript file that "The majority of the known editing events are specific to brain
tissues (Fig. S1)". The finding that brain-specific editing is different from that in non-brain
regions was also reported and highlighted by the previous study. In terms of the brain region
specificity, we as well as Tan. MH reported the segregated RNA editing patterns of the
cerebellum compare to the other brain regions. Please check our response to your later
comment about this issue (page 13 of this response letter).

*2) Is the method of thresholding for calling variants by RNA only now a field accepted method?*

*If so, a reference should be included in results (there is one in the methods). If not, it would be*
*great to take a handful of case from ROSMAP that also have DNA-seq and show that the*
*variants are genuine for the ones that came out as being of strong interest.*

**Authors' response:**

Thank you for pointing this out. Considering the duplicated message, we have removed the
statement about thresholding from the Results section. In addition, we have described this in
more details in the Method section (page 20 line 447 of the manuscript file). Our method of the
thresholding (total reads ≥ 20 and edited reads ≥ 5) for calling variants by RNA is more
conservative than the recommended filters (Li et al., RNA, 2013) (total reads ≥ 5 and edited
reads ≥ 2). In addition, we used the whole genome sequence (a.k.a. DNA-seq) to filter out those
variants on the DNA level. Please check our detailed description in the Method section (page 20
line 448 of the manuscript file): "We have applied posterior filters to filter out those RNA editing
events with (1) total reads less than 20, and (2) alternative reads less than 5, and (3) frequency
less than 10%, and (4) those overlapping with the DNA variants based on the whole genome
sequencing data across the subjects within ROSMAP, MSBB, and MAYO, where some subjects
do not have the RNA-seq data to be involved in the study."

*3) The distribution of editing events in some of the plots in Fig 4 looks to be bimodal. Is there*
*any relationship to RNA quality / technical effects that produces this bimodality?*

**Authors' response:**

We are assuming you are referring the violin plots in the Fig.3 (the Fig. 4 in the previous
version), where we respectfully disagree with the noteworthy issue of the bimodal distributions.
Only 2 out of the total 9 datasets seem to have bimodal distributions (paired ROSMAP DLPFC
and MSBB BM44). We do not think the RNA quality or technical factors are the reason for these
distributions since these are paired samples which means that there are multiple samples from
the same subjects which were processed with the same protocol, at the same time, and by the
same technician. In addition, all of the raw RNA-seq datasets for all 9 datasets were processed
by the same methodologies as described in the Method section and the same quality control
pipeline was applied to all the datasets. In other words, if the RNA quality or the technical
effects were the major reason for the differences, then we should have seen the bimodal
distributions for all the 9 datasets not only 2.

In addition, actually, the violin plots in Fig.3 shows the distributions of the subject-level not the
editing event-level data. We have added more detailed descriptions into the Method section "(1)
Regional comparisons of RNA editing events" (page 24 line 532 of the manuscript file), these
violin plots showed "the distribution of the individual-based overall level of all the called frequent
RNA editing events across brain regions within the same study, which was calculated by
dividing the sum of the % edited reads for all of the RNA editing events by the number of editing
events called within that individual".

*4) What is known about RNA edited mRNAs – do they all make it out of the nucleus? I'm looking*
*at the hugely increased number of calls in the CBE and simply wondering if that is a function of*
*the densely packed nuclei there? So there are more unprocessed nuclear mRNAs sampled in*
*that region than the others?*

**Authors' response:**

Thank you for suggesting that the more densely packed granule cell nuclei in the cerebellum
compared to the other brain regions might act as a potential reason why the MAYO cerebellum
(CBE) dataset has outstandingly greater number of RNA editing events called than the other
datasets. Actually, a previous study also noted a different pattern of RNA editing in the
cerebellum compared to the other brain regions (Tan et al., Nature, 2017). Although RNA
editing can occur in the cell nucleus and cytosol and also within mitochondria, we did observe a
trend that more subjects and more total reads may provide higher probabilities to detect RNA

editing events, which has been discussed in our response to your above comments (page 12
line 481 of this response letter). We show the major picard metrics in the Table S1, where the
MAYO CBE dataset had a significantly greater number of total reads, aligned reads, uniquely
aligned reads, % of ribosome bases and greater median 3' bias than the other datasets. It is
possible that these differences may be due to the fact that cerebellum has more densely packed
nuclei than the other brain regions. However, we do not have the resources to provide solid
evidence for this speculation. We have added more detailed descriptions of the findings in the
Table S1 and highlighted the MAYO CBE dataset on page 5 line 107 of the manuscript file that
"We analyzed the regional differences within each dataset separately because of their
heterogeneities in RNA-seq metrics (Table S1) where the MAYO CBE dataset had a
significantly greater number of greater number of total reads, aligned reads, uniquely aligned
reads, % of ribosome bases and greater median 3' bias than the other datasets."

5) *Could the biological and technical factors adjusted for in the linear models on page 8 be listed*
*in the results section rather than the methods?*

**Authors' response:**

Please check our revised text on page 6 line 127 of the manuscript file that "We conducted
linear mixed models to identify those editing events with a statistically significant difference in
editing levels between 2 brain regions within each study after adjusting for biological (age at
death, sex) and technical confounding factors (postmortem interval and RIN score)."

6) *For protein level quantification of variant harboring proteins (page 10), was the peptide*
*containing the variant removed from overall protein level quant. Leaving it in may affect the*
*overall protein quant. (If it doesn't, it would be great to show that in a supplement).*

**Authors' response:**

No, protein-level quantitation came from the standard Uniprot reference database entries and it
does not incorporate the peptide quantitation of edited variant peptides, which match only to
parts of the 17,112 non-Uniprot protein entries in our custom database. However, in the bottom-
up paradigm of protein assembly and quantitation, for proteins that do have an edited
counterpart, there is a contribution to the signal from the identical peptides of the edited protein.
I.e., variant harboring proteins are quantified by peptides shared with the edited proteoform, so
quantification of total protein is influenced by the presence of edited protein. It is also possible
that some proteins from the custom part of our database were only identified by peptides shared
with standard database entries and chosen randomly to represent the assembled protein from
these peptides, in which case, quantification also represents relative total protein abundance of
unedited RNA-derived protein with an unknown percent contribution from any edited RNA-
derived proteoform that was missed in the database search. To clarify this point, we now state
on page 23 line 502 of the manuscript file that "The protein-level quantitation came from the
standard Uniprot reference database entries and it does not incorporate the peptide quantitation
of edited variant peptides, which match only to parts of the 17,112 non-Uniprot protein entries in
our custom database".

**References:**

- 1. Zhu, D. *et al.* M2 Macrophage Transplantation Ameliorates Cognitive Dysfunction in Amyloid-
beta-Treated Rats Through Regulation of Microglial Polarization. *J Alzheimers Dis* **52**, 483-95
(2016).
- 2. Mathys, H. *et al.* Temporal Tracking of Microglia Activation in Neurodegeneration at Single-Cell
Resolution. *Cell Rep* **21**, 366-380 (2017).

- 3. Readhead, B. *et al.* Multiscale Analysis of Independent Alzheimer's Cohorts Finds Disruption of
Molecular, Genetic, and Clinical Networks by Human Herpesvirus. *Neuron* **99**, 64-82 e7 (2018).
- 4. Tan, M.H. *et al.* Dynamic landscape and regulation of RNA editing in mammals. *Nature* **550**, 249-
254 (2017).
- 5. Ahn, J. & Xiao, X. RASER: reads aligner for SNPs and editing sites of RNA. *Bioinformatics* **31**,
3906-13 (2015).
- 6. Lee, J.H., Ang, J.K. & Xiao, X. Analysis and design of RNA sequencing experiments for identifying
RNA editing and other single-nucleotide variants. *RNA* **19**, 725-32 (2013).
- 7. Wingo, T.S. *et al.* Integrating Next-Generation Genomic Sequencing and Mass Spectrometry To
Estimate Allele-Specific Protein Abundance in Human Brain. *J Proteome Res* **16**, 3336-3347
(2017).

Reviewers' Comments:

Reviewer #1:

Remarks to the Author:

The revised manuscript from Ma et al addresses some of the points raised in the first round of reviews. Although I overall applaud the effort that has gone into this revised version, the manuscript contains still important shortcomings as illustrated below.

1) I appreciate the new analyses that the authors have included to try and dissect the relevance of RNA editing events at gene, transcript and protein level. However, the results described here would be more valuable for the readership of Nature Communications if additional analyses were provided. Specifically, the authors mention that non-coding RNA events have stronger effect than re-coding events on the expression levels of genes and transcripts. Can the authors provide a more detailed analyses of the relationship between location of non-coding RNA editing events (i.e. 5' and 3' UTR regions, intronic transcribed regions) and downstream effects? Is there a general pattern that can be highlighted from such rich datasets? How different RNA editing events impact on the gene and transcript levels, are they generally increased or decreased?

2) This reviewer is not convinced at all by the dismissal of the authors regarding the need for orthogonal validation of statistical associations. It is not clear to me how a statistical association can translate into functional evidence of the role of ORAI2 editing in the accumulation of PHFTau. As for start, the authors do not have any evidence that RNA editing events occurring on ORAI transcript affect the protein level. Although I understand the importance of the statistical analyses supporting a model where perturbation in RNA editing could contribute to the accumulation of Tau pathology, I argue that the model needs to be tested with an orthogonal approach in order to substantiate the claims reported in the manuscript.

Reviewer #2:

Remarks to the Author:

While the authors have done some revision work, it is disappointing that they did not performed analysis on other editing sites, except A-I. I strongly disagree that they do not have statistical power to detect additional changes in editing. I think they are leaving data/analysis on the table which could have been easily done, a recent paper (Tran et al., Nat Neuro, 2018) used far fewer samples (in Autism) and were able to still show robust changes in editing sites in addition to A-I sites.

Since the authors have multi-omic data from the same samples, eg mRNA-expression, splicing changes, RNA-editing, proteomic changes, etc it would be good to know if these changes are occurring in the same patients or not. For example, are those samples/patients showing differential RNA-editing changes same to those showing say differential splicing changes. I think it may be intriguing to know if all these molecular changes are occurring in same AD samples or different samples. They can use PCA analysis to gain insights into this aspect.

Reviewer #3:

Remarks to the Author:

I thank the Authors for their careful consideration of my questions in the initial review. I am satisfied with their responses. This is a novel field and I think getting papers out like this one, which will begin discussions on how to handle these kind of integrative analyses in such rich datasets, is important.

Becky Carlyle, Ph.D.

Instructor in Neurology, Massachusetts General Hospital

REVIEWER COMMENTS

*Reviewer #1 (Remarks to the Author):*

*The revised manuscript from Ma et al addresses some of the points raised in the first*
*round of reviews. Although I overall applaud the effort that has gone into this revised*
*version, the manuscript contains still important shortcomings as illustrated below.*

*1) I appreciate the new analyses that the authors have included to try and dissect the*
*relevance of RNA editing events at gene, transcript and protein level. However, the*
*results described here would be more valuable for the readership of Nature*
*Communications if additional analyses were provided. Specifically, the authors mention*
*that non-coding RNA events have stronger effect than re-coding events on the*
*expression levels of genes and transcripts. Can the authors provide a more detailed*
*analyses of the relationship between location of non-coding RNA editing events (i.e. 5'*
*and 3' UTR regions, intronic transcribed regions) and downstream effects? Is there a*
*general pattern that can be highlighted from such rich datasets? How different RNA*
*editing events impact on the gene and transcript levels, are they generally increased or*
*decreased?*

**Authors' response:**

Following your suggestion, we have conducted a more detailed analyses of the relationship
between locations of the non-coding RNA editing events and downstream effects. We have
added a panel of pie charts to **Figure 4** where we include a pie for each type of non-coding
RNA editing events: those located within 5' UTR (upper left pie), exons (upper right pie), 3' UTR
(lower left pie) and introns (lower right pie). The positive effects were shown in a grade from
pink (non-significance, $P > 0.05$) to brown (nominal-significance, Bonferroni-corrected genome-
wide significance $< P \leq 0.05$) to red (genome-wide significance, $P \leq$ Bonferroni-corrected genome-
wide significance) while the negative effects were shown in a grade from light green (non-
significance, $P > 0.05$) to green (nominal-significance, Bonferroni-corrected genome-wide
significance $< P \leq 0.05$) to dark green (non-significance, $P > 0.05$).

We also added descriptions of these results to the Result section on page 7 line 159 that "A
more detailed analyses of the non-coding events located within different genomic regions
showed that the cis-effects on the expression of the isoforms and proteins were similar. But on
the level of the cis-effects on the expression of the gene mRNA, the intronic non-coding editing
events have less nominally significant effects compared to those events located in the 5'UTR,
exons and 3'UTR (Fig. 4a-c right) where the general patterns of the effect directions /were
positive rather than negative, indicating that the presence of the non-coding RNA editing events
at 5'UTR, exons and 3'UTR were more likely to increase the mRNA expressions of the genes."

*2) This reviewer is not convinced at all by the dismissal of the authors regarding the*
*need for orthogonal validation of statistical associations. It is not clear to me how a*
*statistical association can translate into functional evidence of the role of ORAI2 editing*
*in the accumulation of PHFTau. As for start, the authors do not have any evidence that*

*RNA editing events occurring on ORAI2 transcript affect the protein level. Although I*
*understand the importance of the statistical analyses supporting a model where*
*perturbation in RNA editing could contribute to the accumulation of Tau pathology, I*
*argue that the model needs to be tested with an orthogonal approach in order to*
*substantiate the claims reported in the manuscript.*

**Authors' response:**

We appreciate the reviewer's opinion and have tried to address this comment further in
this version of the manuscript. We note, first, that ORAI2 protein expression
unfortunately is not present in our TMT protein dataset of ROSMAP subjects. Only a
subset of proteins and proteoforms are present in these data (as described in detail in
the original publication [Johnson ECB, Dammer EB., et al., Nat. Med., 2020
May;26(5):769-780]), and the absence of ORAI2 is therefore not informative in and of
itself. It does prevent us from evaluating whether ORAI2 editing affects its own protein
expression. We have added a sentence to highlight this on page 11 line 237: "The
protein level of *ORAI2* was not available, and none of the remaining RNA editing event
showed significant association with their corresponding protein levels."

We have added the analysis of the association between ORAI2 editing event and
protein expression level of TAU, encoded by the *MAPT* gene. With only 78 subjects, we
found a borderline result ($P=0.068$), indicating that ORAI2 editing event might affect the
protein expression level of *MAPT*, consistent with our model. Please check our added
Fig. 6f and statement on page 12 line 267 that "We further found a borderline significant
effect of the ORAI2 editing event on the protein expression of *MAPT* ($P=0.068$) (Fig.
6f)."

Further, we evaluated data from human induced pluripotent stem cell (iPSC) lines differentiated
into neuronal cells with and without expression for *MAPT*, which we have previously shown
leads to Tau phosphorylation (S.E. Sullivan, T.L. Young-Pearse, Brain Res., 2017 Feb 1;
1656:98-106). The data are repurposed from an earlier study (H-U. Klein, C. McCabe, et al.,
Nat. Neuroscience, 2019). Overall, nine *MAPT*-overexpressing and nine control induced
neuronal cell lines were available for analysis from three separate batches. In each batch, each
line was assayed in triplicate. Transcriptome-wide data were generated from each condition.
Unfortunately, *ORAI2* transcripts did not meet our pre-processing parameters in these data
(total reads > 20 and edited reads > 5), so we were not able to evaluate these data to address
the question of whether *MAPT* overexpression caused *ORAI2* expression, which would have
partially addressed the question from the reviewer. Further, this shows that iPSC-derived
neurons do not express *ORAI2* at a meaningful level either at baseline or with perturbation with
Tau, indicating that it is not a relevant context for *ORAI2* over-expression studies. Since we do
not have corroborating evidence at this time, we have elected to remove the mediation analysis
of *ORAI2* from the manuscript. The results of the statistical modeling that we presented in the
previous version of the manuscript are unchanged, but exploring this question further will be
pursued in future efforts.

*Reviewer #2 (Remarks to the Author):*

*While the authors have done some revision work, it is disappointing that they did not*
*performed analysis on other editing sites, except A-I. I strongly disagree that they do not*
*have statistical power to detect additional changes in editing. I think they are leaving*
*data/analysis on the table which could have been easily done, a recent paper (Tran et*
*al., Nat Neuro, 2018) used far fewer samples (in Autism) and were able to still show*
*robust changes in editing sites in addition to A-I sites.*

**Authors' response:**

We have added the analysis for those non A-I editing events. At first, we presented the
distributions of different types of editing events as stacked bars (**Supplementary Fig. S1**) as
shown by Tran et al., Nat Neuro, 2018, and we have added a description of these results to the
Result section (on page 5 line 100): "The majority of RNA editing events are the canonical A-to-I
editing types, which are shown as the A-to-G and T-to-C editing types ($\geq 90\%$) and the C-to-T
and G-to-A types (5%). (**Fig. S1**).

In addition, we added the association analysis for those non-A-to-I editing events (Table S7);
there was no significant associations between the non-A-to-I editing events and available traits,
including clinical diagnosis of Alzheimer's disease, phosphorylated TAU, beta amyloid, neuritic
plaque burden, and cognitive decline. We have added the following statement to the Result
section on page 12 line 274: "There was no significant associations between those non-A-to-I
editing events and traits that we have tested (**Table S7**)".

*Since the authors have multi-omic data from the same samples, eg mRNA-expression,*
*splicing changes, RNA-editing, proteomic changes, etc it would be good to know if*
*these changes are occurring in the same patients or not. For example, are those*
*samples/patients showing differential RNA-editing changes same to those showing say*
*differential splicing changes. I think it may be intriguing to know if all these molecular*
*changes are occurring in same AD samples or different samples. They can use PCA*
*analysis to gain insights into this aspect.*

**Authors' response:**

We have derived 7 principal components based on the top 7 RNA editing events. We
added the statement in the Methodology section on page 26 line 578 that "The scaled
RNA editing levels (%) (mean=0 and SD=1) of each of the top 7 AD-related RNA editing
events were used to derive 7 principal components (PCs) using the R "factoextra" and
"prcomp".

We replicated their associations with the expressions of the genes, isoforms, and
proteins. We presented the results in the **Fig. S7** and added the statement into the
Result section (page 11 line 239): "We derived 7 principal components (PCs) from the
top 7 RNA editing events related to AD. As individual editing events, these PCs were
also showing significant associations with the expressions of genes, isoforms, and
proteins (**Fig. S7**).

*Reviewer #3 (Remarks to the Author):*

*I thank the Authors for their careful consideration of my questions in the initial review. I*
*am satisfied with their responses. This is a novel field and I think getting papers out like*
*this one, which will begin discussions on how to handle these kind of integrative*
*analyses in such rich datasets, is important.*

**Authors' response:**

Thank you for your acknowledgement.

Reviewers' Comments:

Reviewer #2:

Remarks to the Author:

The authors have revised the manuscript to my satisfaction, the manuscript should be accepted without further delay